# Synthetic energy sensor AMPfret deciphers adenylate-dependent AMPK activation mechanism

Martin Pelosse[1,2,3], Cécile Cottet-Rousselle[1], Cécile M. Bidan [4], Aurélie Dupont[4], Kapil Gupta[3], Imre Berger [3] & Uwe Schlattner [1]

AMP-activated protein kinase AMPK senses and regulates cellular energy state. AMPK activation by increasing AMP and ADP concentrations involves a conformational switch within the heterotrimeric complex. This is exploited here for the construction of a synthetic sensor of cellular energetics and allosteric AMPK activation, AMPfret. Based on engineered AMPK fused to fluorescent proteins, the sensor allows direct, real-time readout of the AMPK conformational state by fluorescence resonance energy transfer (FRET). AMPfret faithfully and dynamically reports the binding of AMP and ADP to AMPK γ-CBS sites, competed by $Mg^{2+}$-free ATP. FRET signals correlate with activation of AMPK by allosteric mechanisms and protection from dephosphorylation, attributed here to specific CBS sites, but does not require activation loop phosphorylation. Moreover, AMPfret detects binding of pharmacological compounds to the AMPK α/β-ADaM site enabling activator screening. Cellular assays demonstrate that AMPfret is applicable in vivo for spatiotemporal analysis of energy state and allosteric AMPK activation.

[1] University of Grenoble Alpes and INSERM U1055, Laboratory of Fundamental and Applied Bioenergetics (LBFA) and SFR Environmental and Systems Biology (BEeSy), Rue de la Piscine, Domaine Universitaire, 38610 Gières, France. [2] European Molecular Biology Laboratory, 71 Avenue des Martyrs, 38042 Grenoble CEDEX 9, France. [3] Bristol Synthetic Biology Centre BrisSynBio, Biomedical Sciences, University of Bristol, 1 Tankard's Close, Bristol BS8 1TD, UK. [4] University of Grenoble Alpes, CNRS, Laboratoire Interdisciplinaire de Physique (LIPhy), 140 Rue de la Physique, 38402 Saint-Martin-d'Hères, France. Correspondence and requests for materials should be addressed to I.B. (email: imre.berger@bristol.ac.uk) or to U.S. (email: uwe.schlattner@univ-grenoble-alpes.fr)

M aintenance of energy homeostasis in the body is a vital prerequisite for endergonic cellular processes. To maximally exploit the free energy of ATP hydrolysis, the ratio of ATP to ADP must be kept at a high level. AMP-activated protein kinase (AMPK) is an evolutionary conserved heterotrimeric complex capable of sensing and responding to changes in cellular energy state[1–3]. AMPK is activated by multiple parallel and potentially synergistic pathways. However, many of the underlying molecular mechanisms remain elusive.

In vivo, phosphorylation of T172 in the catalytic α-subunit is required to activate AMPK, predominantly by liver kinase B1 (LKB1)[4,5], but in certain cells types also by calcium/calmodulin-dependent protein kinase 2 (CaMKK2 or CaMKKβ)[6–8], counteracted by a range of phosphatases[9,10]. Importantly, when cellular ATP is depleted due to imbalanced production and consumption, AMP and ADP levels increase and competitively replace ATP at up to two of the four cystathionine beta synthase (CBS) sites, CBS1 and CBS3[11–13]. Pairwise, these CBS sites form two Bateman domains in the regulatory γ-subunit. CBS4 is likely bound constitutively to AMP in vivo although it can be exchangeable in vitro[14], while CBS2 remains unoccupied[9]. AMP acts by direct allosteric activation of AMPK, while both AMP and ADP promote α-T172 phosphorylation and inhibit dephosphorylation by phosphatases[13]. Most direct pharmacological activators of AMPK, including A-769662 or compound 991, and likely also a yet to be identified intracellular metabolite, bind to the allosteric drug and metabolite (ADaM) site at the α/β-interface[15,16]. Allosteric activation by the ADaM, CBS1, and CBS3 sites appears to be additive[17] and, at least in vitro, sufficient for AMPK activation even in absence of α-T172 phosphorylation[16].

Each of these activation mechanisms requires cross-talk between the catalytic α and the regulatory β- and/or γ-subunits. This cross-talk involves a conformational switch which we first observed by small angle X-ray scattering (SAXS) in full-length AMPK[18]. Subsequent electron microscopy and X-ray crystallographic studies with truncated heterotrimer confirmed this switch[14,19], revealing an α regulatory subunit-interacting motif (αRIM) directly contacting CBS3 in the γ subunit[19–21]. More recently, solution studies using hydrogen/deuterium exchange mass spectroscopy (HDX-MS)[22,23] or luminescence energy transfer[24] provided insight into CBS site contributions to AMP- and ADP-induced conformational changes. Again, activator binding to the ADaM site induces rearrangements between α- and β-subunits, involving the capping of α-KD by β-CBM[24–26]. Once activated, AMPK relieves energy stress by triggering a large variety of cell-type-specific responses slowing ATP consumption while accelerating ATP synthesis, acting on metabolic pathways, signaling cascades, and gene expression[9,11,27]. Beyond its central role in energy homeostasis, AMPK also regulates cell cycle, shape, motility, proliferation, autophagy, apoptosis, and hypothalamic appetite control[28]. Due to these manifold functions, AMPK became a highly attractive pharmacological target for instance for treating type II diabetes and obesity[29,30].

Here, we set out to harness the adenylate-induced conformational switch to create a genetically encoded metabolic biosensor capable of reporting cellular energy states. Our sensor, AMPfret, relies on FRET occurring between fluorescent proteins (FPs) fused to suitable AMPK subunit termini as deduced by combinatorics. AMPfret faithfully reports on conformational changes upon binding of allosteric activators, relevant for AMPK activation and description of cellular energy state. These changes are readily reversible upon inactivation, in contrast to existing FRET sensors depending on fluorescent AMPK substrates[31–33]. We use our biosensor AMPfret to reveal mechanisms of AMPK activation in vitro, and to detect allosteric AMPK activation and energy stress in living cells.

## Results

**AMPfret design and engineering.** AMPfret converts the AMP-induced conformational change into a measurable signal by exploiting FRET between two FPs. Based on highly AMP-responsive α2β2γ1 AMPK[34], we first fused cyan FP (CFP) and yellow FP (YFP), respectively, to all combinations of N- or C-termini of two different AMPK subunits using the ACEMBL system[35] (Table 1, Supplementary Fig. 1a, b). Wild-type AMPK activity was preserved and measurable FRET was observed in most constructs (Table 1, Supplementary Figs. 1d, 2). Two fusion constructs exhibited a FRET change of sufficient magnitude upon AMP binding to serve as a sensor. These two AMPK variants shared a CFP-tag at the C-terminus of α2, while the YFP-tag was either at the C-terminus of γ1 (AMPfret 1.0) or β2 (AMPfret pre-2.0) (Supplementary Fig. 3a, b). The observed increase in FRET suggests a more compact conformation of AMP-bound AMPK, consistent with conformational rearrangements in structural studies[14,18,22,36]. In all constructs, FRET values correlated well with FP distances estimated from published AMPK structures (PDB IDs: 5ISO and 4CFF)[26] (Table 1).

We further engineered AMPfret to maximize adenylate-induced FRET changes for cellular applications. We removed presumably flexible low-complexity regions at subunit termini, and added instead a rigid 8-amino acid α-helix linker between α2 C-terminus and mseCFPΔ11. Meant to amplify molecular movements of AMPK, this indeed doubled FRET signal change upon AMP binding (Supplementary Fig. 3c). AMPfret constructs retained endogenous AMPK affinities for AMP and ADP, as demonstrated for AMPfret 1.0 (Supplementary Figs. 4, 5). Importantly, AMPfret 2.0 also responded to compound 991, an activator targeting the ADaM site (Suppl. Figure 3c). To further optimize the sensor, we selected the more efficient mseCFPΔ11/cpVenus FRET pair[37] resulting in AMPfret 1.1 and AMPfret 2.1 sensors (Fig. 1, Table 1).

**Adenylate-dependent rearrangements and allosteric activation.** Our AMPfret biosensors faithfully report changes in AMP concentration occurring already below 10 μM and faster than the 10 s temporal resolution (Fig. 1d)[38]. The concentration dependence of this sensitive and rapid response by fluorometry confirmed saturation below 15 μM and yielded an affinity constant below 2 μM (Fig. 2a, Supplementary Fig. 4b, e), corroborating that AMPfret senses physiologically relevant AMP concentrations. Next, we tested ADP, which is likewise an indicator of cellular energy depletion and a known AMPK activator. In a physiological concentration range, FRET measurements yielded an affinity constant of 7.5 μM for ADP (Supplementary Fig. 4c, f). FRET changes induced by AMP were stable over a wide range of pH and salt conditions (Supplementary Fig. 4g, h) and not detected if GMP was used (Fig. 2b). AMP-induced FRET changes also correlated well with AMP-dependent allosteric activation of AMP-fret, as determined with CaMKKβ-pre-activated sensor and acetyl-CoA carboxylase (ACC) as kinase substrate (Fig. 2a, red trace). A slight shift of AMPfret activity towards higher AMP concentrations in the kinase assay is likely due to the presence of ATP (competing with AMP) and higher temperature (affecting CBS binding constants; Supplementary Fig. 4i). We conclude that AMPfret retains wild-type affinity for adenylates and responds to their fluctuations within a physiological range. Moreover, our sensor also reports allosteric activation of phosphorylated AMPfret by AMP.

**T172 phosphorylation and AMP-dependent rearrangements.** We investigated the impact of AMPK activation loop phosphorylation on AMP-dependent FRET signal change (Fig. 2c).

**Table 1 Overview of AMPfret constructs tested for AMP responses**

| AMPfret construct | Organization | YFP peak[a] | FRET ratio[b] | Distances (Å)[c] | FRET change[d] |
|---|---|---|---|---|---|
| AMPK 221 | $\alpha_2 \_ \beta_2 \_ \gamma_1$ | / | / | / | / |
| AMPfret A (AMPfret pre-2.0) | $\alpha_2$-CFP $\_ \beta_2$-YFP $\_ \gamma_1$ | ++ | 1.444 | 13.8 | + |
| AMPfret B | CFP-$\alpha_2 \_ \beta_2$-YFP $\_ \gamma_1$ | ± | 0.492 | 78.9 | − |
| AMPfret C (AMPfret 1.0) | $\alpha_2$-CFP $\_ \beta_2 \_ \gamma_1$-YFP | + | 0.639 | 59.8 | + |
| AMPfret D | $\alpha_2$-CFP $\_ \beta_2 \_$ YFP-$\gamma_1$ | ± | 0.513 | 62.7 | − |
| AMPfret E | CFP-$\alpha_2 \_ \beta_2 \_ \gamma_1$-YFP | − | / | 78.5 | − |
| AMPfret F (AMPfret CTL) | CFP-$\alpha_2 \_ \beta_2 \_$ YFP-$\gamma_1$ | − | / | 72.7 | − |
| AMPfret G | $\alpha_2 \_ \beta_2$-YFP $\_ \gamma_1$-CFP | ± | 0.524 | 50.0 | − |
| AMPfret H | $\alpha_2 \_$ YFP-$\beta_2 \_ \gamma_1$-CFP | ± | 0.464 | (63.3) | − |
| AMPfret I | $\alpha_2 \_ \beta_2$-YFP $\_$ CFP-$\gamma_1$ | + | 0.637 | 53.1 | − |
| AMPfret J | $\alpha_2 \_$ YFP-$\beta_2 \_$ CFP-$\gamma_1$ | ± | 0.559 | (58.2) | − |
| AMPfret K | $\alpha_2$-CFP $\_$ YFP-$\beta_2 \_ \gamma_1$ | + | 0.648 | (67.8) | ± |
| AMPfret L | CFP-$\alpha_2 \_$ YFP-$\beta_2 \_ \gamma_1$ | − | / | (17.7) | − |
| AMPfret 2.0 | $\alpha_2$-helix-CFP $\_ \beta_2$-YFP $\_ \gamma_1$ | ++ | 2.032 | 19.0 ± 11.6 | ++ |
| AMPfret 1.1 | $\alpha_2$-mseCFP$_{\Delta 11} \_ \beta_2 \_ \gamma_1$-cpVenus | + | 0.727 | 55.0 | + |
| AMPfret 2.1 | $\alpha_2$-helix-mseCFP$_{\Delta 11} \_ \beta_2$-cpVenus $\_ \gamma_1$ | ++ | 2.026 | 19.0 ± 11.6 | ++ |

[a]YFP peak visibility annotated as: ++ strong, + detectable, ± shoulder, − absent
[b]FRET ratio (YFP/CFP) at baseline
[c]Distances between tagged termini determined from PDB IDs: 5ISO and 4CFF26. In brackets: distances involving the β N-ter lacking in these structures. For AMPfret 2.0/2.1: a range is given, since orientation of the added α-helix linker could not be predicted
[d]Change of FRET ratio upon AMP addition annotated as: ++ strong, + average, ± detectable, − non-detectable

The critical activation loop residue T172 was either phosphorylated by CaMKKβ, or replaced with a phospho-mimetic (T172D) or non-phosphorylatable mutation (T172A). None of these alterations caused any effect on the AMP-induced FRET response. Even in a physiological setup of 3 mM MgATP competing with AMP, both T172D and T172A mutants reacted similarly to AMP. Thus, AMP binding to the γ-subunit and the resulting conformational changes are independent of phosphorylation of the activation loop.

**AMPfret enables AMPK activator screening**. We asked whether AMPfret can report allosteric activation by small molecule AMPK activators A-769662 (20 μM) or compound 991 (2 μM) occurring at the ADaM site. AMPfret 1.1 and even more so AMPfret 2.1 exhibited pronounced FRET increase with 991, while the response to A-769662 was insignificant (Fig. 2d). These results are consistent with the ADaM site involving the β-subunit (tagged in AMPfret 2.1), and 991 exhibiting high affinity to the β2 subunit (present in all AMPfret sensors). In contrast, A-769662 preferentially binds β1-AMPK[27]. Re-screening of our initial constructs for their response to 991 confirmed that those most responsive to AMP are also most responsive to 991 (Supplementary Table 1), suggesting that both allosteric activators trigger similar activating AMPK conformations. We further investigated the effect of A-769662 in presence or absence of AMP to account for potential cumulative effects (Fig. 2e). In the presence of 20 μM AMP, A-769662 increased the FRET ratio in a concentration-dependent manner, supporting an additive effect of A-769662 and AMP for AMPK activation, consistent with previous reports[39,40]. Indirect AMPK activators inhibiting mitochondrial respiration, such as metformin[41,42] did not trigger a FRET change. Also, local conformational changes induced in the α-subunit kinase domain by the kinase inhibitor staurosporine did not translate into an activating conformation of the entire heterotrimeric complex as detected by AMPfret (Fig. 2d). Nicotinamide adenine dinucleotides (NADs) were reported to bind to CBS sites[20] and directly activate AMPK in case of NADH[43], although this has been subject to debate[44]. With AMPfret, even NADs concentrations exceeding physiological[45,46] failed to trigger FRET signal change, taking into account contaminating AMP in the NAD preparation (Supplementary Fig. 6). We thus postulate that NADs do neither induce any activating AMPK conformation in cells in vivo. In summary, AMPfret biosensor can be used for screening AMPK activators interacting at the γ-CBS sites and also at the α/β ADaM site.

**Functional role of CBS sites**. AMPfret allows direct monitoring of conformational changes that have been linked to allosteric activation by AMP and protection from dephosphorylation by ADP and AMP, in absence of ATP normally required for kinase assays. This enabled us to directly analyze the individual roles of AMP and ADP for the three adenylate-binding CBS sites by site-specific mutations (CBS1: L128D+V129D, CBS3: V275G+L276G, CBS4: S315P)[14] that preserved kinase activity (Supplementary Figs. 1e, 5a–d). Our experiments clearly confirmed CBS3 as the main site of conformational response to saturating concentrations of both AMP and ADP (Fig. 3a)[14,47]. Mutation of CBS1 only diminished the FRET response to AMP but did not affect the ADP response. Surprisingly, an inverse situation was observed when mutating CBS4, which abolished the FRET response triggered by ADP, but not by AMP.

We subjected all CBS mutants to relevant AMPK activity assays using immunodetection of P-ACC and P-AMPK or to ³²P incorporation into ACC (Fig. 3b–d, Supplementary Fig. 5e–h). Allosteric activation by increasing amounts of AMP was about twofold in unmutated AMPfret and CBS4 mutant, but much reduced in CBS1 mutant, and entirely lacking in CBS3 mutant. In fact, activation of CBS1 mutant was biphasic, without activation below 5 μM AMP, and reduced activation at higher concentrations. These data are in excellent agreement with observed AMP-induced FRET changes. Protection from dephosphorylation of T172-phosphorylated AMPfret in presence of PP2Cα and increasing amounts of AMP or ADP was observed in unmutated AMPfret and again absent in CBS3 mutant. In contrast to what we observed with allosteric activation, CBS1 mutant showed here characteristics of unmutated sensor. Interestingly, CBS4 mutation did not affect protection by AMP, but only blunted protection by ADP - again in excellent agreement with the FRET data. Our results thus indicate that CBS3 is essential for conformational changes that lead to allosteric activation and protection from dephosphorylation. CBS1 appears to be necessary for full allosteric activation, but is dispensable for protection from

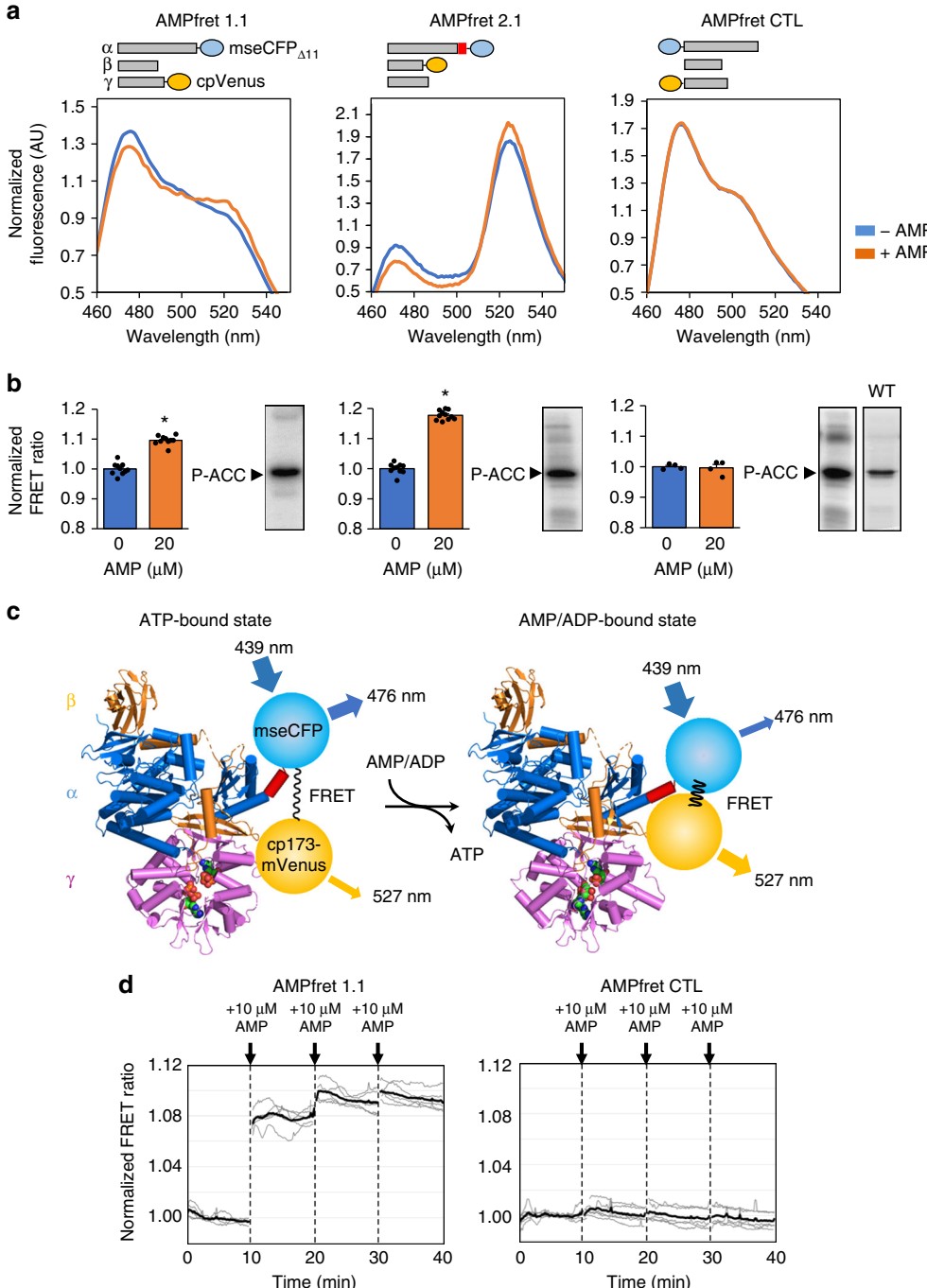

**Fig. 1** FRET changes in response to AMP. **a** AMP-dependent variation of fluorescence emission spectra of AMPfret constructs 1.1 and 2.1 and AMPfret-negative control (CTL) (blue line: no AMP, orange line: 20 μM AMP). Spectra normalized to the isosbestic point at 512 nm. AMPfret topologies are depicted on top (bars: AMPK subunits, red box: putatively rigid helix, blue circles: cyan fluorescent protein mseCFP$_{\Delta 11}$, yellow circles: yellow fluorescent protein cp173-mVenus). **b** Normalized FRET ratio of AMPfret constructs calculated from **a** (same color code), and corresponding autoradiograms of in vitro kinase activity assays using acetyl-CoA carboxylase (ACC) as substrate. Data and error bars represent mean ± SEM ($n \geq 7$, *$p < 0.001$, Student's $t$-test). **c** Schematic drawing of AMPfret 2.1 operating mode. Within an α2β2γ1 AMPK heterotrimer (α: blue, β: orange, γ: pink), a CFP variant (blue circle) fused to a small rigid α-helix linker (red) was added at the α2-C-terminus, while a YFP variant (yellow circle) was added at the β2-C-terminus. Binding of AMP (and ADP) to the γ-subunit induces a conformational change which modifies distance and/or orientation between the fluorophore couple, thus increasing the non-radiative energy transfer (FRET; black wave) between them. Excitation/emission wavelengths are indicated. The model is based on AMPK core structures co-crystalized with either ATP (PDB ID: 4EAK) or AMP (PDB ID: 4EAI)[16] and extended for missing parts by the full-length AMPK structure (PDB ID: 5ISO), thus visualizing some of the conformational changes (AMPK in schematic backbone representation, ATP and AMP in space filling representation). **d** Kinetics experiments of AMPfret 1.1 (left) and AMPfret control (CTL, right) responses to the addition of 10 μM AMP: the ratiometric FRET signals were measured every 10 s, corrected for cross-talk (CT = 0.20) and direct excitation (DE = 0.04) and normalized by the average signal before addition of AMP. Gray and black lines represent individual samples and their mean, respectively ($n = 5$)

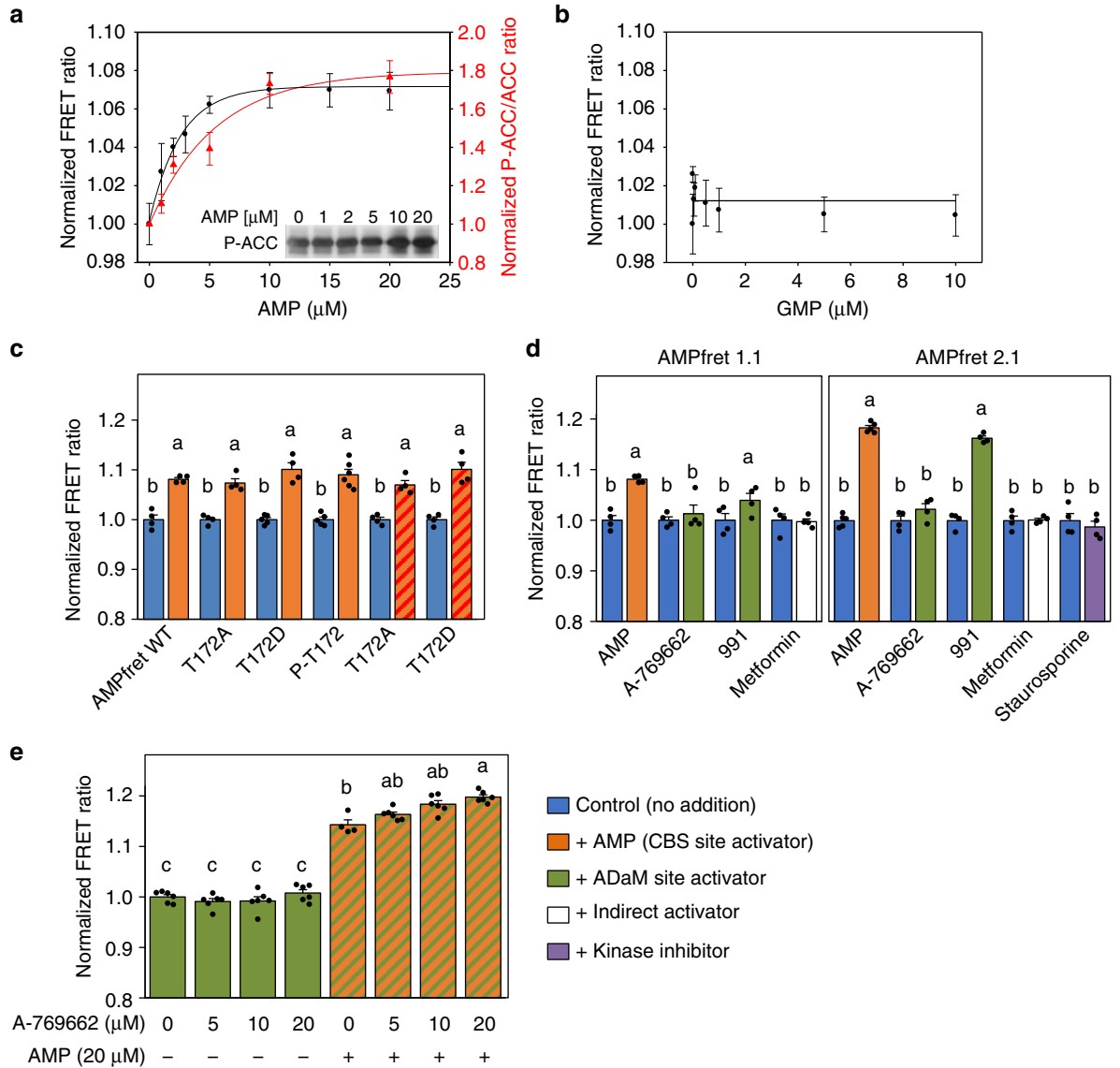

**Fig. 2** FRET changes report AMPK allosteric activation. **a** AMP-concentration dependence of FRET (black) compared to AMP-induced allosteric activation (red) of AMPfret 1.0. This activation was determined with pre-activated AMPK in presence of 200 μM ATP, followed by immunoblotting for phosphorylated acetyl-CoA carboxylase (ACC; insert); quantified band intensities were normalized to control at 0 μM AMP. Data and error bars represent mean ± SEM ($n \geq 3$). Note: ATP in the kinase assay shifts the curve to higher AMP concentrations. **b** GMP-concentration dependence of FRET, showing absence of FRET change. Points represent mean ± SEM ($n \geq 3$). **c** Phosphorylation of T172 does not interfere with AMP-induced conformational changes reported by FRET. AMPfret 1.0 wild type (WT), pre-phosphorylated (P-T172), or mutated (T172A/D) proteins were incubated in absence (blue) or in presence of 20 μM AMP (orange), or with additional 3 mM $Mg^{2+}$-complexed ATP (orange dashed red). Note: FRET changes are independent of MgATP. Data and error bars represent mean ± SEM ($n \geq 3$). **d** AMPfret response to different AMPK activators. AMPfret 1.1 (left) or 2.1 (right) were incubated in absence (blue) or in presence of AMP (20 μM, orange), A-769662 (20 μM, green), 991 (2 μM, green), metformin (500 μM, white) or staurosporine (25 μM, purple). Note: no FRET change is induced by the indirect activator metformin or the protein kinase inhibitor staurosporine. Data and error bars represent mean ± SEM ($n \geq 4$). **e** Additive effect of allosteric activators bound to CBS (AMP) and ADaM sites (A-769662). FRET ratio of AMPfret 2.0 incubated with A-769662 (5–20 μM) in absence (green) or presence of 20 μM AMP (green dashed orange). Note: a concentration-dependent effect of A-769662 is only significant in presence of AMP. Data and error bars represent mean ± SEM ($n \geq 3$). All FRET signals are normalized to control (no adenylates). After checking normality and equality of variance, statistical significance was analysed by two-way ANOVA followed by Bonferroni multiple comparison. Means sharing the same letter do not differ significantly ($p \leq 0.02$)

dephosphorylation. Of note, our experiments show that CBS4, previously described as a non-exchangeable site[26], is apparently not involved in AMP-triggered allosteric activation or protection mechanisms but may be required for ADP-mediated protection.

**AMPK regulation by free ATP and adenylate mixtures.** How AMPK can sense low micromolar concentration of AMP in presence of a millimolar concentration of ATP, a difference of three orders of magnitude, remains a profound riddle. It was proposed that only non-complexed (free) ATP but not MgATP competes with AMP[48–50]. Indeed, different affinities for ATP as compared with MgATP were determined for CBS sites[23]. AMP-fret enabled us, to directly test this hypothesis, since AMPfret readout does not require ATP (Fig. 4a). The baseline FRET signal and the FRET increase induced by 20 μM AMP or 200 μM ADP

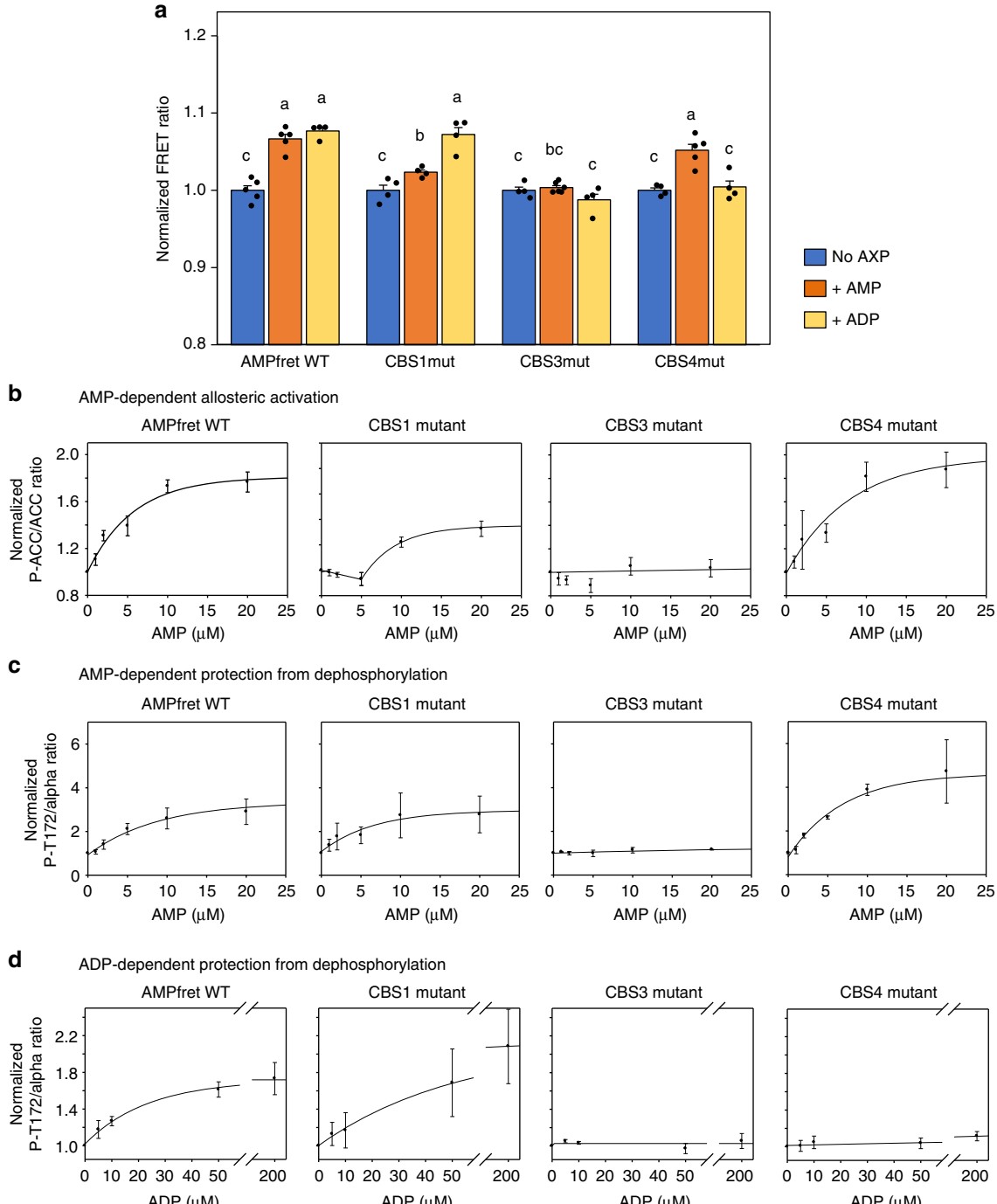

**Fig. 3** Deciphering allosteric activation and protection against dephosphorylation. **a** FRET ratios of AMPfret 1.0 wild type (WT) and corresponding CBS site mutants in presence of AMP (30 μM, orange bars) or ADP (200 μM, yellow bars) were normalized to the adenylate-free control (blue bars). Data and error bars represent mean ± SEM ($n \geq 5$). **b** AMP-induced allosteric activation of AMPfret WT and its CBS site mutants. AMPK activity was quantified with increasing concentrations of AMP and fixed MgATP (200 μM) by immunoblotting for phosphorylated acetyl-CoA carboxylase (P-ACC; see Supplementary Fig. 5) and data normalized to kinase activity in absence of AMP. Data and error bars represent mean ± SEM ($n = 3$). **c, d** AMP- and ADP-dependent protection against dephosphorylation of AMPfret WT and its CBS site mutants. AMPK activity was quantified with pre-phosphorylated AMPK, phosphatase PP2Cα and increasing concentrations of AMP or ADP by immunoblotting for α-subunit P-T172 (see Supplementary Fig. 5) and data normalized to dephosphorylation in absence of ADP. Data and error bars represent mean ± SEM ($n = 3$). After checking normality and equality of variance, statistical significance was analysed by two-way ANOVA followed by Student–Newman–Keuls multiple comparison. Means sharing the same letter do not differ significantly ($p \leq 0.02$)

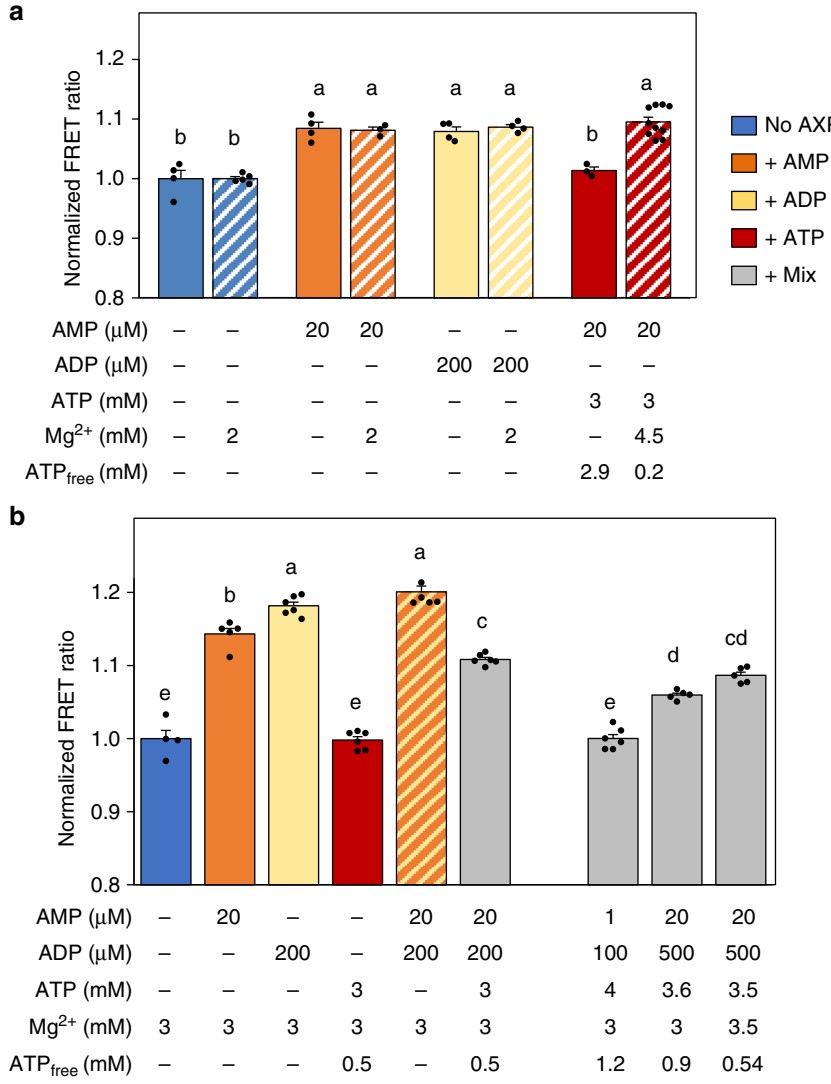

**Fig. 4** Effects of adenylate mixtures on FRET changes. **a** Free ATP, not $Mg^{2+}$-complexed ATP, competes with AMP. $MgCl_2$ does not affect AMPfret 1.0 baseline FRET (blue bars) and FRET induced by AMP (orange) or ADP (yellow), but prevents ATP-induced FRET inhibition (red). Plain and dashed bars, respectively, correspond to FRET signal measured in absence or in presence of $Mg^{2+}$, respectively. **b** AMPfret 2.1 responses to individual adenylates and adenylate mixtures as indicated below the bars: no adenylates (blue), AMP (orange), ADP (yellow), ATP (red) AMP plus ADP (orange dashed yellow), mixtures of AMP, ADP, and ATP (gray). Bars to the right represent physiological mixtures with a shift to energy deprivation and emphasize the role of free ATP. The FRET signals are normalized to control (no adenylates). All data and error bars represent mean ± SEM ($n \geq 3$). After checking normality and equality of variance, statistical significance was analysed by one-way ANOVA followed by Bonferroni multiple comparison. Means sharing the same letter do not differ significantly ($p \leq 0.02$)

were not sensitive to excess $Mg^{2+}$ in our experiments, in agreement with structural studies performed with AMP in the presence or absence of $Mg^{2+}$ ions[51]. Complementation of 20 µM AMP with 3 mM ATP entirely reversed the FRET signal to baseline, clearly showing competition of free ATP with AMP for inducing conformational changes. However, when excess $Mg^{2+}$ was added, the resulting MgATP had no detectable effect, indicating that MgATP did not compete with AMP. These results compellingly demonstrate that only free ATP but not MgATP competes with AMP for binding to CBS sites causing conformational changes, rationalizing the mechanism of AMPK sensing fluctuations in ATP, ADP, and AMP levels despite their vastly different cellular concentrations.

Using most sensitive AMPfret 2.1, we analyzed in more detail how physiological mixtures of adenylates affect AMPK conformational changes (Fig. 4b). In presence of 3 mM $Mg^{2+}$, addition of 20 µM AMP or 200 µM ADP increased the FRET

signal, while 3 mM ATP (yielding 0.5 mM free ATP) did not, as expected. Adding AMP and ADP in combination led to a maximal FRET increase of about 20%. Further addition of ATP into this incubation mixture clearly reduced the FRET signal. Next, approximating physiological conditions, we choose a baseline resting state of 1 µM AMP, 100 µM ADP and 4 mM ATP (Fig. 4b), i.e. lower AMP and ADP concentrations as used earlier for unstressed conditions[23]. This mixture did not alter the baseline FRET signal, thus demonstrating that free ATP (1.2 mM in this case) could entirely outcompete AMP and ADP at the CBS binding sites under these conditions. To simulate cellular energy stress, where ATP consumption exceeds ATP generation and the adenylate kinase reaction adjusts new adenylate equilibria, we increased AMP (20-fold) and ADP (fivefold), and decreased ATP (by 0.9-fold), thus decreasing free ATP (by 0.75-fold). Under such conditions, the FRET signal increased by about 7%. Since free ATP seems to be a decisive parameter for AMPK activation, but

its true physiological value is difficult to estimate, we further varied ATP concentrations while keeping AMP and ADP concentrations constant. With a decrease of free ATP to about 0.5 mM, the FRET ratio further increased to about 10% above resting state. Our data lead to important conclusions. First, also in physiological adenylate mixtures, the monitored conformational changes rely on a binding equilibrium between free ATP and AMP or ADP at the AMPK CBS sites, leading to dynamic regulation of AMPK activation. Moreover, energy stress-related adenylate mixtures trigger FRET changes up to about 10%, a value to be expected when deploying AMPfret 2.1 into living cells, and which should allow relative quantification of adenylates or adenylate ratios.

**AMPfret sensor in living cells.** To deploy AMPfret 2.1 in living cells, we pasted AMPfret genes into our MultiMam system[52] for mammalian cell transfection. HEK293t, 3T3-L1, and HeLa were transiently transfected with AMPfret 2.1 to observe FP signal (Fig. 5). AMPfret was detected mainly in the cytoplasm of these cell lines (Fig. 5a, e, h), providing proof-of-concept evidence for the spatial resolution of AMPfret signals and suggesting that compartment-specific detection is feasible using our sensor. Nuclear AMPfret fluorescence was present, but not sufficient to reliably calculate FRET (Supplementary Fig. 7). We speculate that a putative nuclear localization signal in α2 may be obstructed by the FP in the tagged construct, while a nuclear export signal

identified in the α-C-terminal part remains active[53]. Next, we supplied cells with 1 mM AICAR to allosterically activate AMPK, and recorded AMPfret emission spectra over time (Fig. 5b). In all three cell lines tested, half-maximal response occurred during the first 5–6 min, reaching saturation at about 30 min (Fig. 5c, f, i). With an AMPfret response to AMP occurring faster than 10 s (Fig. 1d), this kinetics probably reflects cellular AICAR uptake and conversion into the AMP-analogue ZMP. Maximal FRET signals (4–5% for HEK293T and HeLa, ~7% for 3T3-L1) and their variability differed slightly among the cell types. This may reflect cell-type-specific differences, or different AMP/ATP ratios at basal conditions to which the curve was normalized. If the latter was the case, then higher AMP/ATP ratios at baseline would diminish the exploitable FRET signal range for subsequent experiments. Under the same conditions, we separately determined AMPK activation in 3T3-L1 cells by detection of phosphorylated ACC (Fig. 5g). The excellent correlation between FRET signal and AMPK activation kinetics in 3T3-L1 underscores that AMPfret can indeed report allosteric AMPK activation in living cells. To further explore whether AMPfret can be used for detection of both allosteric AMPK activation and spatio-temporal analysis of endogenous adenylate ratios, we inhibited glycolysis in HEK293t cells (Fig. 5c) by using 2-deoxyglucose (2-DG, 3.3 g/L) at low glucose (1 g/L) as compared to normal glucose (4.5 g/L). Again, AMPfret responded by an increase in the FRET signal, but much slower as compared to AICAR, with a half-

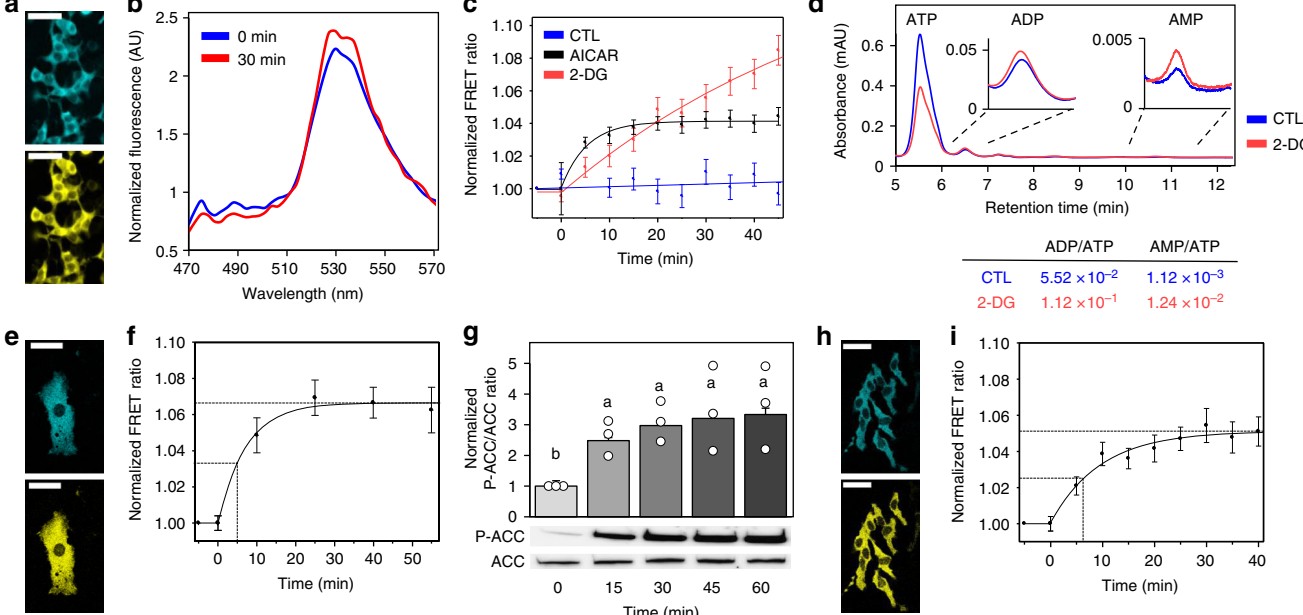

**Fig. 5** AMPfret in cellulo. **a–d** HEK293T cells transfected with AMPfret 2.1. **a** Micrograph of CFP and YFP channels (scale bars: 50 μm). **b** Fluorescence emission spectra before (0 min, blue) and after addition (30 min, red) of 1 mM AICAR, showing increase at 527 nm (cpVenus) and decrease at 478 nm (mseCFP$_{\Delta 11}$) over 30 min. **c** Normalized AMPfret FRET signal variation (YFP fluorescence at 530 ± 5 nm/CFP fluorescence at 478 ± 5 nm) after incubation with 1 mM AICAR (black), 20 mM 2-DG (red) or complete medium (control CTL, blue). Data of cell population normalized to values before treatment. Data and error bars represent mean ± SEM ($n = 64$, $n = 27$, and $n = 39$ cells from at least two independent experiments for 2-DG, AICAR, and control, respectively). **d** HPLC quantification of adenylates in HEK293T cells treated for 50 min with complete medium (control CTL, blue) or 20 mM 2-DG medium (red). Curve (average of three-independent experiments) showing peaks with typical retention times of 5.5 min (ATP), 6.5 min (ADP) and 10.8 min (AMP), and adenylate ratios calculated thereof (same color code). **e–g** 3T3-L1 cells transfected with AMPfret 2.1. **e** Micrograph of CFP and YFP channels (scale bars: 50 μm). **f** Normalized AMPfret FRET signal variation after incubation with 1 mM AICAR. Data of cell population normalized to values before AICAR addition. Data and error bars represent mean ± SEM ($n = 9$ cells of two-independent experiments). **g** AICAR-dependent activation of AMPK verified by parallel immunoblots for P-ACC and ACC (lower panel) and their quantification (upper panel). Data and error bars represent mean ± SEM ($n = 3$). After checking normality and equality of variance, statistical significance was analysed by one-way ANOVA followed by Bonferroni multiple comparison. Means sharing the same letter do not differ significantly ($p \leq 0.02$). **h, i** HeLa cells transfected with AMPfret 2.1. **h** Micrograph of CFP and YFP channels (scale bars: 50 μm). **i** Normalized AMPfret FRET signal variation after incubation with 1 mM AICAR. Data of cell population normalized to values before AICAR addition. Data and error bars represent mean ± SEM ($n = 51$ cells of three-independent experiments)

maximal response occurring only after ~25 min At the end of the 45 min observation period, the FRET signal was already about twice as high as with AICAR, without yet reaching saturation. These data are consistent with ongoing energy depletion induced by 2-DG, slowly decreasing ATP and increasing ADP and AMP levels, which then translate into allosteric AMPK activation. Direct quantification of adenylates by HPLC confirmed that the AMPfret signal 50 min after 2-DG addition corresponds to a 2- and 10-fold increase of ADP/ATP and AMP/ATP ratios, respectively (Fig. 5d). Different to 2-DG, AICAR uptake directly generates the allosteric activators ZMP and AMP, but does not alter ADP and ATP levels, probably causing the more rapid activation, but liming the maximal AMPfret signal. These data provide compelling evidence that AMPfret indeed affords readout of endogenous adenylate ratios. Of note, AMPfret readout primarily depends on adenylate concentrations and their fluctuation, independent of AMPK T172 phosphorylation. Since phosphorylation is required for full kinase activation, AMPfret reports allosteric activation in cells only if basal T172 phosphorylation is afforded by active AMPK kinases.

## Discussion

Freely accessible pools of cytosolic ATP, ADP and AMP constitute the major source of free energy driving metabolism, and they represent the most fundamental readout to describe changes in cellular energy state[54]. Biochemical or NMR-based measurements in whole cells and tissues or extracts have limits, since they do not account for both compartmentalized and structurally bound adenylates, or fail to detect small alterations. In principle, genetically encoded fluorescent reporters[55] can be engineered to provide information on physiologically relevant, local adenylate concentrations[31,37,56]. Here we exploit the naturally evolved adenylate affinities of the key metabolic regulator AMPK[57] to create our genetically encoded synthetic sensor, AMPfret. Using a matrix approach for positioning FRET reporter pairs, we identified and optimized two distinct AMPfret constructs exhibiting exploitable AMP-dependent FRET changes. AMPfret faithfully reports conformational changes within the full-length, fully functional AMPK heterotrimer linked to altered AMP/ATP and ADP/ATP ratios sensed at the CBS sites, and the resulting AMPK activation in vitro and in living cells. AMPfret is also sensitive to pharmacological activators such as compound 991 binding at the ADaM site. Thus, AMPfret allows mechanistic studies of AMPK activation, monitoring cellular energy state, and identification of AMPK activators. It further provides direct evidence for the conformational switch model of AMPK activation that we proposed earlier based on SAXS with wild-type AMPK heterotrimer[18], later confirmed by structural and biochemical studies[14,21,36,58] which used truncated AMPK subunits[14], or indirect in vitro isotope or chemical tagging approaches[22–24].

AMPfret constructs that produced suitable FRET signal changes are tagged exclusively at the C-termini of either α2 and γ1 (AMPfret 1.0/1.1) or α2 and β2 (AMPfret 2.0/2.1), which imply that these termini are moving significantly relative to each other upon AMP or ADP binding. HDX-MS analysis also revealed singular changes close to the C-termini of β2 and γ1, while in α2 changes appeared to occur mostly within the active site[22,23]. None of our β2/γ1-tagged constructs show significant AMP-induced FRET variation, suggesting that these subunits do not move significantly relative to each other. This does not exclude structural rearrangements in the core of the β- and γ-subunits[24] or a movement of both tagged termini as a rigid body relative to the α-C-terminus. Likewise, no AMP-induced FRET variation is seen with the N-terminally tagged α2/γ1-pair, although distance variations were reported between α1/γ1-N-

termini[24]. Likely, movements between α/γ- C-termini are much more important.

AMPfret retains wild-type AMPK properties, including kinase activity, regulation by phosphorylation, and native adenylate affinities that evolved to sense critical adenylate changes at the onset of cellular energy stress[44]. Nuclear import of AMPfret is reduced, but still occurring. The FRET readout directly correlates with sensor-bound adenylates, thus AMPfret reports physiologically relevant increase in AMP/ATP and ADP/ATP ratios. This provides a useful tool to detect, with good temporal resolution, cellular energy states, and AMP- or ADP-dependent AMPK activation, i.e. allosteric activation and protection of dephosphorylation. FRET changes occur independent of AMPK αT172 phosphorylation, with both γ-CBS and α/β-ADaM site activators, and thus already report an increased potential for AMPK activation in absence of upstream kinases. This may be an advantage in cellular models that lack sufficient upstream kinase activity. However, although canonical AMPK activation requires αT172 phosphorylation, there is increasing evidence for an allostery-only activation by the sole additive or synergistic action of CBS and ADaM site activators[16,40,59]. Although the achieved activity may remain lower in cells as compared to in vitro assays[60], this opens new avenues for AMPK activating treatments, and AMPfret is able to report such activation. All these properties unique to our AMPfret tool set it apart from known AMPK activity reporters based on artificial, fluorescent AMPK substrates[31,33]. Those do not report the contribution of altered cellular energy state and are limited in their temporal resolution for detecting AMPK activation, and in particular inactivation (which would depend on separate phosphatase activities). An important advantage of AMPfret for in vitro studies is its activation-related FRET. In contrast to classical kinase activity assays, it does not require ATP, which is prone to non-enzymatic degradation to ADP and AMP[44,61], all interfering with binding at the CBS sites. Thus, AMPfret represents a powerful tool to analyze adenylates with respect to the different activation mechanisms, as we demonstrated here. AMPfret reports binding of AMP (in absence of ATP) at one or more binding sites in a range up to 20 μM and with an affinity of about 1.5 μM, close to expected values[20,34,44]. The two exchangeable CBS sites[20] either have similar affinity for AMP, or the difference cannot be resolved by fitting the AMPfret concentration-response curve. Similar to AMP, ADP induces a FRET signal indicating a single affinity (in absence of ATP) in the range of about 7.5 μM. For both adenylates, competition with free ATP in vivo would shift the binding curve to higher AMP concentrations as seen in the kinase activity assays, in very good agreement with physiological fluctuations of these adenylates[20,62,63]. Thus, AMPfret affinities likely describe the true responsiveness of AMPK to AMP and ADP, in contrast to some published affinities appearing either too high (in the sub-micromolar range)[24] or too low[23]. AMPfret also enabled us to closely analyse the influence of non-complexed (free) as compared to $Mg^{2+}$-complexed ATP. Our data compellingly demonstrate that not only for CBS binding[23,34], but also for induced conformational changes the true competitor of AMP (and ADP) is non-complexed ATP, which is at least 10-times less abundant in cells as compared to MgATP, while there is no effect of $Mg^{2+}$ on AMP and ADP. This rationalizes how micromolar AMP concentrations can outcompete 1000-times higher total ATP concentrations for AMPK activation[20,64]. Importantly, neither of the two forms of ATP induced a FRET signal on their own, challenging the notion that ATP can serve as an active allosteric modulator[24]. However, the fact that we do not see ATP-induced movements in our AMPfret sensors characterized here does not rule out the possibility that ATP-induced conformational changes may exist, requiring further future study.

Mutating the different adenylate-binding CBS sites within AMPfret led to key insight into allosteric regulation by adenylates at the mechanistic level of induced conformational changes. CBS3 appears to be essential for conformational alterations by both AMP and ADP, involving allosteric activation (by AMP) and protection from dephosphorylation (by AMP and ADP). This is in agreement with mutational[14,23] and structural findings[25,26] which showed that CBS3 harboring several naturally occurring mutants involved in cardiac pathologies (e.g. R531G/Q)[65,66] can transmit conformational changes to the catalytic α-subunit involving a presumably unstructured loop region comprising the α-RIM1 and α-RIM2 motifs interacting with CBS2 and CBS3, respectively[19,21]. Mutation at CBS1 reduces AMP-dependent conformational changes and attenuates AMP allosteric activation, in particular by shifting it to higher concentrations. This suggests that AMP-binding to CBS1 is required for full allosteric activation, consistent with earlier models[19,24]. ADP-dependent conformational changes were unaltered by CBS1 mutation, suggesting that this site is not involved in protection from dephosphorylation. CBS4 can exchange nucleotides under extreme conditions, and is typically occupied by AMP or ADP also in recombinantly produced complex[14]. Occupation of this site was reported to increase AMP binding at CBS3[23], but CBS4 mutation in AMPfret did not show any significant effect on FRET and allosteric activation. In contrast, we found that mutating S315 at CBS4 fully abolished ADP-induced changes. Since we see this effect specifically only with ADP and not with AMP, an indirect effect of S315P on CBS3 affinities is unlikely.

Consistent with the notion that the active state of AMPK adopts a unique, compact conformation with many conserved features[20,21,25,26], AMPfret reports movements of the same subunit termini with similar amplitude for AMP and ADP. However, they also unveil hitherto unknown differences in AMP- and ADP-dependent conformational changes, since each involves a specific set of CBS sites. Indeed, since ADP acts through protection of the α subunit from dephosphorylation[13] and not via allosteric activation, induced conformational changes must differ at the molecular level from those induced by AMP. These differences may be linked to the manner α-RIM1 and α-RIM2 motifs connect the kinase domain with the CBS sites[19,21].

In vivo, ADP and AMP signals would occur together, with ATP consumption immediately increasing ADP, and - via the adenylate kinase reaction - also AMP[9,67]. Indeed, our data are consistent with both adenylates being important for AMPK activation in vivo, with activating conformational changes occurring at physiologically relevant concentrations, in particular very low micromolar AMP, and by using physiological adenylate mixtures. Although AMPfret was not designed for monitoring binding of direct AMPK activators to the ADaM site at the α/β interface[25,26,68], AMPfret 2.1 (α2/β2-pair FP-tagged) revealed increased FRET when incubated with 991, in contrast to A-769662, consistent with the lower affinity of the β2 isoform for this activator[15,69]. The conformational changes reported by AMPfret here are likely different to those triggered by AMP or ADP binding to CBS sites, but apparently result in a similar reduction in the distance between α2/β2-C-termini. This fact, together with the almost identical outcome of our construct screening with AMP and 991, may reflect overall similar intramolecular and intermolecular rearrangements within the AMPK heterotrimer, irrespective of the initial activator or binding site involved, as also suggested earlier[23]. These are different from conformational changes triggered by binding at the kinase domain active site, as staurosporine does not induce an AMPfret signal.

CBS and ADaM sites can cooperate in allosteric activation to a degree that activates AMPK even when not phosphorylated at α-T172 in vitro[16,40,59]. Such synergy between activated CBS and ADaM sites induces an overall conformation close to α-T172-phosphorylated AMPK[60] which we could determine in vitro by observing similar responses of AMPfret 2.1 to AMP and compound 991. Although with currently available synthetic activators such allostery-only activation seems to be limited in human cells[60], there is significant potential here for future pharmacological development. Indeed, AMPK modulators are much sought after to treat metabolic diseases[70]. With its design and conserved native features of AMPK activation, AMPfret is a powerful tool to screen and identify direct activators binding to γ-CBS and α/β-ADaM sites. AMPfret does not report binding of antagonistic competitors like ATP, kinase substrates or inhibitors, or indirect activators like metformin. This could be advantageous for identifying the elusive endogenous metabolite binding at the ADaM site, which may provide allosteric AMPK activation independent of energy state or even T172 phosphorylation[27,59]. Further, AMPfret can likely be engineered to distinguish between ADaM site activators with preference for β1 or β2 subunits.

AMPfret 2.1 reports AICAR- and 2-DG-induced FRET signals in different cell lines minutes after addition, very likely directly following AICAR conversion into the AMP-analogue ZMP or inhibition of glycolysis, respectively. The different time course and magnitude of the AMPfret response to AICAR and 2-DG illustrate how AMPfret reports changes in endogenous adenylate ratios and thus cellular energy state. As shown here in 3T3-L1 cells, AMPfret monitors adenylate levels and allosteric AMPK activation without apparent lag phase. This sets AMPfret apart from existing AMPK activity reporters based on artificial fluorescent substrates that require about an hour to detect AICAR effects[31]. Since the AMPfret signal is directly proportional to the activation-linked conformational change, it should also rapidly reverse upon decreasing AMP or ADP levels, as it does in vitro when adding ATP. This would allow the analysis of transient AMPK activation, difficult to achieve with available AMPK substrate reporters with reversibility depending on phosphatases. Finally, since AMPfret reports conformational changes in absence of T172 phosphorylation, it can even be used in cell lines lacking significant upstream kinase activity, like HeLa cells[33].

An advantage of genetically encoded protein-based biosensors is that, in principle, any parameter can be fine-tuned by molecular engineering. During the present work, we have successfully improved FRET amplitudes by introducing more advanced FPs and by increasing the signal range by ~100% with the insertion of a small, rigid α-helix. The resulting 20% range in the AMPfret 2.1 FRET signal in vitro may still be limiting for certain applications in living cells. It is conceivable that further optimization of AMPfret to improve the FRET range by engineering could provide an even more powerful sensor. Since AMPfret is a reporter of AMPK allosteric activation that occurs under conditions of metabolic stress and that can be engaged by synthetic ligands, experiments can be envisioned to identify potent AMPK modulators. Importantly, the flexible design of our AMPfret constructs facilitates switching subunit isoforms to screen for isoform-specific activators that could have tissue-specific effects[71], and screening of endogenous metabolites that allosterically activate at the ADaM site. We anticipate many exciting roles for AMPfret in a wide range of applications, as a highly versatile tool to accelerate discovery.

## Methods

**Molecular cloning**. AMPK subunits used in this work are the AMPK α2 catalytic subunit from *Rattus norvegicus* (*PRKAA2*; Gene ID: 78975), the AMPK β2 subunit from *Homo sapiens* (*PRKAB2*; Gene ID: 5565) and the γ1-subunit isoform from *Rattus norvegicus* (*PRKAG1*; Gene ID: 25520). All AMPfret constructs carry an N-terminal deca-histidine tag and a protease cleavage site for tobacco etch virus

(TEV) NIa protease on the sequence coding for α2. In the first generation of AMPfret biosensors (including AMPfret 1.0 and 2.0), ECFP and EYFP were fused to tag the aforementioned AMPK subunits by sequence and ligation-independent cloning method (SLIC)[72] or conventional restriction cloning (see Supplementary Table 2 for primer list). Through a matrix approach, made possible by the versatility of the MultiColi expression system[35], all combinations of full-length AMPK heterotrimer tagged on two of its three subunits with either a CFP variant or a YFP variant at the N- or C-terminus were created. α2-, β2-, and γ1-derived AMPfret subunits were, respectively, cloned in pACE1, pDC and pDS of the MultiColi expression system. For the most recent AMPfret generation (AMPfret 1.1 and 2.1) we used the mseCFPΔ172V/cp173Venus FRET pair. Mutations at the α2T172 (T172D and T172A) and at three γ1 CBS sites shown to abrogate adenylate binding[16,23] (CBS1: L128D + V129D, CBS3: V275G + L276G and CBS4: S315P), truncations and linker engineering at the interface of AMPK subunit and the FP were realized by Self-SLIC[73]. Briefly, it consisted in amplifying the whole backbone of a coding vector using primers that contain the desired mutation plus an overlapping region, which allowed the self-recircularization after T4 DNA polymerase exonuclease treatment. False-positive products are avoided by a digest of the PCR template using Dpn1. Identification of mutated clones was facilitated by the introduction of new restriction sites without changing the amino acid sequence, taking advantage of the redundancy of the genetic code. Expression vectors were obtained through Cre-LoxP recombination of the single subunit containing plasmids. For in cellulo work, AMPfret 2.1 subunits were transferred by SLIC into plasmids of the MultiMam expression system (α2-mseCFPΔ11, β2 and γ1-cp173Venus, respectively, in pACEmam2, pMDS and pMDK)[52]. Positive clones were identified through extensive restriction digest pattern analysis and all constructs were verified by DNA sequencing.

**Protein engineering.** Two strategies were applied to amplify the AMP-induced FRET ratio of AMPfret. First, we removed non-folded residues located at the termini of AMPK subunits and fluorescent proteins. Residues were identified through the examination of X-ray structure available at that time (PDB IDs: 2Y94 and 5ISO[20]) and secondary structure prediction. Second, based on observations by Sivaramakrishnan et al. that ER/K amino acids repeat forms a rigid α-helix[74], we positioned such motifs (2- to 12-AA length) between the α-subunit and CFP. Only one α-helix linker was inserted to fix one fluorophore, thus avoiding to increase distance between both of them.

**Expression and purification.** All versions of AMPfret were expressed in BL21 (DE3) Star cells (Invitrogen) and protein expression was carried out overnight at 18 °C in autoinducing medium[75]. Cells were collected by centrifugation and flash frozen in liquid nitrogen. For purification, bacterial cell pellets were resuspended in Lysis buffer (0.5 M sucrose, 30% glycerol, 50 mM Tris, pH 8, 100 mM NaCl, 2 mM MgCl₂, 2 mM β-mercaptoethanol, 20 mM imidazole and protease inhibitors). Next, bacterial cells were lysed by sonication and cell-free extract obtained by centrifugation (60 min, 75,000 g, JLA 25.50 rotor) was applied on Ni-NTA Superflow resin (Qiagen). Resin was then washed with lysis buffer, wash buffer (50 mM Tris, pH 8, 100 mM NaCl, 20 mM imidazole, 2 mM MgCl₂, 2 mM β-mercaptoethanol) and high salt buffer (wash buffer + 1 M NaCl). Proteins were eluted by applying elution buffer (wash buffer + 400 mM imidazole). Imidazole was removed through an overnight dialysis in buffer A (50 mM Tris, pH 8, 100 mM NaCl, 2 mM MgCl₂, 2 mM β-mercaptoethanol). AMPfret protein complexes were applied onto a QXL column (GE Healthcare) to remove nucleic acids and non-stochiometric AMPK complexes. AMPfret complexes were eluted using a 100 mL continuous gradient of buffer B (50 mM Tris, pH 8, 1 M NaCl, 2 mM MgCl₂, 2 mM β-mercaptoethanol) and were collected at a salt concentration of ~200 mM NaCl. For maximal purification, AMPfret constructs were applied onto a Superose 6 10/300 gel filtration column (GE Healthcare) preequilibrated with SEC buffer (50 mM Tris, pH 8, 200 mM NaCl, 2 mM MgCl₂, 2 mM β-mercaptoethanol, 5 mM spermidine). AMPfret eluted at a volume corresponding to globular proteins of a ~280 kDa molecular weight, suggesting a rather elongated shape of AMPfret. After adding glycerol to a final concentration of 50%, the purified AMPfret constructs were placed at −20 °C until use for further experiments. Purity was assessed at each purification step through SDS-PAGE analysis.

**Enzymatic kinase assay.** AMPfret constructs and AMPK 221WT (3 pmol) were activated by incubation with purified CaMKKβ (1 pmol) for 20 min at 30 °C in kinase buffer (200 μM ATP, 40 μM AMP, 5 mM MgCl₂, 1 mM DTT, and 10 mM HEPES pH 7.4). Purified GST-ACC fragment (200 pmol) was incubated for 20 min at 37 °C in presence of pre-activated AMPfret constructs in kinase buffer containing 200 μM [γ−32P]-ATP (specific activity 650 mCi/mmol ATP). Reaction mixtures were then separated on SDS-PAGE gel. Specific AMPfret activities were revealed using a Typhoon imaging system (GE Healthcare). AMP-dependent allosteric activation was evaluated under the same conditions, except that various concentrations of AMP were used. Enzymatic kinetics of AMPfret constructs and their CBS site mutants were performed as above in presence or absence of AMP (20 μM) in the kinase buffer, except that less AMPfret sensor was used (200 fmol). To stop the reaction at given time points, protein loading dye was added to the reaction mixture and immediately heated to 98 °C for 3 min AMPK

kinase activity was also revealed by immunoblotting for P-Ser79-ACC and total ACC (see immunoblots) using unlabeled ATP. ADP-dependent protection against dephosphorylation was assessed using AMPfret constructs pre-phosphorylated by CaMKKβ and repurified over a Superose 6 10/300 column. Then, AMPfret (50 ng) was incubated with PP2Cα (200 ng; Sigma) for 2 h at 37 °C in presence of various amounts of ADP (0–200 μM). The phosphorylation status of AMPfret constructs was evaluated by immunoblotting for P-T172 α-subunit and total α-subunit (see immunoblots). The amount of PP2Cα to use and other conditions were determined through preliminary tests.

**Fluorometric FRET assay.** FRET signal variation in presence of different compounds (nucleotides, chemicals, ions) was measured using a fluorometer (Photon Technology International). Excitation wavelength was set to 430 nm, and emission spectra were recorded from 450 to 600 nm with a step size of 1 nm and an integration time of 0.2 s. AMPfret constructs (15 pmol) were incubated in a quartz cuvette (Hellma) in a final volume of 150 μL (Spectro buffer: 50 mM Tris, pH 8, 200 mM NaCl, 5 mM MgCl₂, 2 mM β-mercaptoethanol). Effects of nucleotides and other compounds (prepared in Spectro buffer) were assessed by comparing FRET ratios (ratio of emission maxima at $527 \pm 2$ nm and at $476 \pm 2$ nm) in their presence or absence. Mg²⁺ effects on FRET were investigated with variable concentrations of Mg²⁺ (0–10 mM) and AMPfret stock solution diluted to yield final Mg²⁺ concentrations below 20 μM. Once acquired using the Felix software, spectra were treated under Excel or SigmaPlot 13.0. In concentration series, data were fitted with Sigma Plot 13.0 to single site binding kinetics. Ratios between Mg²⁺ and free ATP and the derived amount of Mg²⁺-ATP were calculated using Maxchelator software at http://maxchelator.stanford.edu at the given experimental conditions.

**Ratiometric FRET kinetics.** To estimate the kinetics of the response of the AMPfret biosensors at 10 s resolution, we used an epifluorescence microscope (Olympus IX83). The changes in ratiometric FRET signal were monitored in a volume of 200 μL of AMPfret 1.1 or negative control AMPfret CTL (both 20 pmoles) in Spectro buffer. AMP concentration was successively increased by addition of 1 μL of 2 mM AMP stock solution every 10 min, i.e. 10 μM steps. Every 10 s, the solution was alternatively excited with 442 nm and 515 nm light (Fianium supercontinuum laser) and both CFP (donor) and YFP (acceptor) signals were simultaneously acquired on a sCMOS camera (Hamamatsu ORCA-Flash 4.0 v2). Practically the two emission channels (donor and acceptor) were split with a dichroic mirror (510LP, F48–510, AHF Technologies), filtered (519LP F37–519 and 475/50 F39–477, AHF Technologies, Chroma) and spatially shifted to fit simultaneously on the camera chip. This alternating-laser excitation method (ALEX)[38] enabled us to obtain the FRET ratio corrected for cross-talk (CT) and direct excitation (DE) from the formula,

$$\text{FRET ratio} = \frac{I_{DA} - CT.I_{DD} - DE.I_{AA}}{I_{DD}}, \tag{1}$$

where $I_{DA}$ is the average intensity collected in the acceptor channel upon donor excitation (442 nm), $I_{DD}$ is the average intensity collected in the donor channel upon donor excitation, $I_{AA}$ is the average intensity collected in the acceptor channel upon acceptor excitation (515 nm). The theoretical values of the correction factors $CT = 0.20$ and $DE = 0.04$ were calculated from the specifications of the filters used and the fluorophore spectra. We normalized the FRET ratios for each experiment to the average basal value calculated over the 10 min before the first addition of AMP in the solution, and finally averaged the five experiments. The calculation was done with Matlab software (MathWorks).

**Cell culture and transfection and treatments.** 3T3-L1 (ATCC, CL-173) and HeLa cells (ATCC, a gift from Prof. Peter Cullen, Univ. of Bristol, UK) were cultured in Dulbecco Modified Eagle Medium (DMEM, Institut de Biotechnologies Jacques Boy) containing 4.5 g/L glucose, 10% SVF (Pan-Biotech, lot P291905), 10 mM HEPES (PAA cell culture company), penicillin and streptomycin (Gibco) as well as non-essential amino acids (Sigma). HEK293T cells (ATCC, CRL-3216) were cultured in DMEM (Gibco 41965039) containing 4.5 g/L glucose, 10% SVF, penicillin and streptomycin. For transfection of cells at ~80% confluence, medium was replaced by OptiMEM (Gibco) and MultiMam-derived AMPfret 2.1 coding plasmid was transfected using either Lipofectamine 2000 (Invitrogen) for 5–6 h (3T3-L1 and HeLa cells) or PolyFect transfection reagent (Qiagen) following manufacturer recommendations (HEK293T cells). For extracts, cells were grown to ~80% confluence in Ø35 mm dishes before replacing the medium by either fresh complete medium (control), complete medium with AICAR (1 mM final), or 2-DG medium (containing 3.3 g/L 2-DG, 1 g/L glucose instead of 4.5 g/L glucose), and cells grown for variable time until lysis. For AMPfret analysis, cells cultured in 8-well Labtek (Nunc) in complete medium was observed by the confocal microscopy. Medium was then replaced by complete medium containing AICAR (1 mM final) or 2-DG (3.3 g/L 2-DG and 1 g/L glucose) without moving the 8-well plates.

**Confocal microscopy.** AMPfret transfected cells were observed with a Leica TCS SP2 or SP8, both equipped with an incubation chamber (37 °C, 5% CO₂) and a Plan Apo ×63/1.40 oil or a HCX plan Apo ×20/0.70 dry as objectives. The instrument's

autofocus system was used (LCS SP8) or z-stacks (4 tiles distributed through 10 μm) were acquired and combined (LCS SP2) to avoid focal drift during the experiments and to tolerate cell shape remodeling. For excitation, the 458 nm Argon laser (<20% power) was used and pictures were taken at 512 × 512 or 1024 × 1024 pixel resolution with a 400 Hz scanning frequency and 4 lines averaging in the CFP (478 ± 5 nm) and the YFP (530 ± 5 nm) channels every 5 min for 40–60 min using HyD (LCS SP8) or PMT (LSC SP2) detectors. After averaging of z-stacks intensities and background subtraction (rolling ball radius: 200 pixels) using FIJI, both CFP and YFP fluorescence intensity values were extracted from the recorded pictures using FIJI. FRET signal variations for single cells were calculated from averaged fluorescence intensities (YFP fluorescence value/CFP fluorescence value).

**Protein-free extracts and HPLC nucleotide analysis.** HEK293T cells were put on ice and washed three times with PBS before lysis and metabolic quenching with 0.6N perchloric acid for 1 min Lysate and cell debris were then scraped, centrifuged (2 min, 13,000 rpm, 4 °C), supernatants neutralized with 2 N KOH, 0.3 M MOPS and again centrifuged (10 min, 13,000 rpm, 4 °C). In these supernatants and in solutions of commercial ATP, NADH, and NADPH (Roche), adenylates were quantified by HPLC (Varian with Proplus autosampler 410; Agilent Polaris C18 stationary phase; 60% $CH_3CN$, 40% $H_2O$ mobile phase; detection at 254 nm) using calibration curves established for each nucleotide.

**Protein extracts and immunoblots.** 3T3-L1 cells were flash-frozen in liquid nitrogen and put on ice. 200 μL of buffer containing 50 mM Tris, pH 8, 200 mM NaCl, 2 mM β-mercaptoethanol were added and cells were scratched and then frozen in liquid nitrogen. After thawing, cells were sonicated 5 s. Lysates were clarified by centrifugation and supernatant was kept. Proteins were quantified using the Bradford reagent and identical protein amounts were loaded onto 7.5% SDS-PAGE gels. Proteins transferred on a nitrocellulose membrane were probed with antibodies (all rabbit, dilution 1:1000, Cell Signaling) against total ACC (#3662), P-Ser79-ACC (#3661), total AMPKα (#2532), or P172-AMPKα (#2535). Specific bands were visualized using a LAS 4000 imager (GE Healthcare) and P-ACC and ACC bands were quantified using ImageJ.

**Statistics.** Data are presented as mean ± standard error of the mean (SEM) and statistically analyzed by Sigma Plot 11.0 (Systat Software Inc., San José, CA, USA). Data were checked for normality (Shapiro–Wilk) and equal variance. If not stated otherwise, data were then analyzed for significance by one-way or two-way ANOVA, depending on the experimental design, followed by post hoc multiple pairwise comparison by either Bonferroni or Student–Newman–Keuls methods. Differences were considered significant when $p \leq 0.02$. In the figures, means sharing the same letter do not differ significantly.

**Reporting summary.** Further information on experimental design is available in the Nature Research Reporting Summary linked to this article.

## Data availability

Data supporting the findings of this manuscript are available from the corresponding authors upon reasonable request. A reporting summary for this Article is available as a Supplementary Information file. The source data underlying Figs. 1b, 2a, 5g and Supplementary Figs. 1b-e, 3b, 6e-h are provided as a Source Data file.

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

## Acknowledgements

The authors thank all members of the Schlattner and Berger labs for discussion, Dr. Małgorzata Tokarska-Schlattner and Stéphane Attia for statistical analysis and adenylate determination, Prof. Jia-Wei Wu (School of Life Sciences, Tsinghua University, Beijing, China) for suggestions during an early phase of the project, and Dr. Alexei Grichine (IAB, UGA, Grenoble, France) for advice on live cell imaging. This work was mainly supported by the French regional government of Rhône-Alpes with a *CIBLE* grant and the French National Research Agency in the framework of the *Investissements d'Avenir* program (ANR-15-IDEX-02). This work received also funding from the European Commission (EC) in Framework Programme (FP) 6 (LSHM-CT-2004–005272, EXGENESIS) and FP 7 (KBBE-2013-613879, SynSignal) in an early phase. Further support came from the SFR BEeSy, a federal research structure at the University Grenoble Alpes, and BrisSynBio, a BBSRC/EPSRC Research Centre for synthetic biology at the University of Bristol (BB/L01386X/1).

## Author contributions

U.S., M.P. and I.B. conceived the study with input from C.C.-R., C.B. and A.D. M.P. carried out most experimental work with input from I.B. and U.S. C.C.-R. and C.B. performed some confocal and in vitro FRET analyses, respectively. K.G. expressed and purified some reagents. U.S., M.P., I.B. and A.D. designed experiments and interpreted data. M.P., I.B. and U.S. wrote the manuscript together with input from all authors.

## Additional information

**Competing interests:** M.P., I.B. and U.S. declare competing interest and are inventors on a patent application describing AMPfret applications. All the remaining authors declare no competing interests.

