## [Peer Review File · Nature Communications]

Reviewers' Comments:

Reviewer #1:

Remarks to the Author:

The authors have made use of the conformational change that occurs on binding of AMP or ADP to AMPK (first detected by their group by SAXS in 2008) to develop a FRET sensor based on the alpha-2/beta-2/gamma-1 complex of AMPK. This allows the conformational change that occurs upon binding of AMP or ADP to be detected and monitored, independently of changes in phosphorylation of Thr172 and hence overall kinase activity. Most of the manuscript describes the use of these sensors in cell-free assays, and this has generated some useful information such as: (i) confirmation of previous proposals that only free ATP, rather than the MgATP complex, competes with AMP or ADP for binding to the gamma subunit; (ii) confirmation that previously reported effects of NAD can be accounted for by contamination with AMP; (iii) analysis of the contributions of the individual gamma subunit sites to the binding of AMP and ADP [although see point (4) below]. However, although the FRET sensor is also tested in intact cells, the statement that the authors themselves make on lines 441-442 ("The resulting 20% range in the AMPfret 2.1 FRET signal in vitro may still be limiting for certain applications in living cells") seems to indicate that the relevance of the current generation of sensors to intact cell studies may be somewhat limited. Nevertheless, this manuscript does describe a potentially useful addition to the armory of experimental tools used to study the AMPK system and, with attention to the points raised below, would be of interest to the AMPK aficionados.

MAJOR POINTS:

1. Fig. 2a, and elsewhere: the authors should consider the use of semilog plots (i.e. log₁₀ [AMP] versus FRET ratio), which are often better at displaying differences in responses, such as the rightward shift when the output was measured using p-ACC:ACC in the presence of ATP.
2. Line 236: can the authors state why they think that 1 μM AMP, 100 μM ADP and 4 mM ATP approximate to physiological conditions?
3. Line 263: can the authors state what they mean by the nuclear localization signal in alpha-2? The authors should discuss Kazgan et al (2010) Mol Biol Cell 21: 3433, which defined a nuclear export sequence (NES) at the C-terminus of alpha-2 that is also conserved in alpha-1. How has the NES in alpha-2 been affected by the construction of their FRET sensors, especially the second generation?
4. Fig. 5: although it is reasonable that they should initially assess the efficacy of their sensor in intact cells using AICAR, the real challenge is whether it can detect changes in the ratios of endogenous nucleotides when compounds that disturb cellular energy balance are added. Why did they not attempt this?
5. Discussion section: the previous paper that is perhaps most relevant to the discussion of their results is Gu et al (2017). Although it used different experimental approaches to address the roles of the three CBS sites, some of the mutations used to eliminate binding to the individual sites even appear to have been the same as those used in this paper. It would be helpful for the Discussion section of this manuscript to include a more direct comparison of the results of Gu et al (2017) with those of the present authors, as well as their interpretation (which in some cases appears to be conflicting).

MINOR POINTS:

6. Line 60: whether AMP exchanges with ATP at two of the four CBS sites remains the subject of debate. Gu et al (2017) proposed instead that the CBS1 site may normally always have ATP bound. It might be better to state: "AMP and ADP levels increase and competitively replace ATP at up to two of the four cystathionine beta synthase (CBS) domains".

7. Line 62: It might be more accurate to describe the CBS2 site as “unoccupied” rather than “inactive”.

8. Line 93: what exactly do they mean by “common FRET sensors”? Perhaps “existing FRET sensors” is a better description?

9. Line 98: they had already described in the Introduction how the conformational change that occurs upon AMP binding to AMPK was discovered and defined, and I would suggest that the words “we had discovered” are superfluous here and should be deleted.

10. Line 187: although it was possible to identify which residues in gamma-1 were mutated in Supplementary Fig. 6a-c, to find out what they were replaced with it was necessary to dig down into the Methods section. The authors should please provide details of the mutations that were made in a much more accessible fashion.

Reviewer #2:

Remarks to the Author:

The article by Pelosse et al describes an interesting and clever tool where the authors use a genetically encoded FRET system to report on ligand-induced conformational changes in the multisubunit energy sensing kinase AMPK. Based on many prior studies, AMPK is known to be a dynamic and flexible complex and the authors use their system, AMPfret, to monitor FRET changes in response to binding of known ligands. Of particular interest is the ability to use AMPfret in live cells to get a kinetic read of AMPK engagement, and the ability to carefully titrate nucleotide ratios in vitro (including +/- Mg) to interrogate AMPK response.

Major points:

1) Although this paper provides many interesting insights, a number of questions remain regarding interpretation and utility of this AMPfret system. First, it is interesting to note that AMPfret yields in increase in FRET signal upon both nucleotide binding to the gamma subunit, and allosteric activator binding to the distal ADaM site. The authors suggest that this may reflect similar rearrangements in AMPK, irrespective of the initial activator or binding site involved – and this observation leads to the proposal that AMPfret can be used as a novel screen for AMPK activators. It would be interesting and important to demonstrate if a similar FRET change occurs upon inhibitor binding to the ATP site. A potent and available tool compound known to inhibit AMPK is staurosporine. If staurosporine binding also leads to an increase in FRET, then this may require that changes be made to the discussion regarding interpretation of the nature of the FRET signal – and may demonstrate that this system reports generally on ligand binding rather than reporting on an ‘active’ conformations.

2) In results, paragraph 2, line 119, the authors describe a construct, AMPfret 2.0, that was biochemically characterized to confirm it retained endogenous affinity for AMP – yet it appears that this confirmation was made using FRET. It would be better to use an independent technique – aside from the FRET system being developed in this paper – to validate the protein FRET construct. An independent binding assay would be useful to build confidence that the FRET signal of these constructs correlates with affinity.

3) Discussion in the manuscript suggests that AMPfret can be used as a cellular sensor of energetics, and data from Figure 5 shows an exciting kinetic trace for cells treated with AICAR. My question and concern is regarding discussion of how this can be used. Although treatment with AICAR shows a 7% increase in signal for AMPfret, I don’t know how to interpret this in terms of cellular energetic state. Can this truly be used to assess the concentrations of different adenylates as suggested? What would these values be for the experiment shown? If the authors truly propose that this probe can be used to monitor cellular energy state (line 299), then it would be nice to see at least some rough quantitation from in-cell studies (in addition to the normalized kinetics

shown).

4) Regarding CBS mutants, unless I misinterpreted Supplementary Figure 1 and 6, CBS3 mutant appears significantly functionally impaired both in the absence and presence of AMP. This seems counter to the text in lines 186 and 187 and makes interpretation of data concerning this mutant uncertain.

5) Given the panel of AMPfret constructs described, it would have been interesting to see if some of the constructs that were non-FRET-responsive to AMP were instead responsive to 991 or staurosporine. This could reveal selective sensors for different binding sites and would increase the potential breadth, utility, and impact of these AMPfret sensors.

Minor comment:

1) In Supplementary Figure 6E, it would be helpful to directly label the blue and orange lines as +/- AMP in the actual figure rather than simply the figure legend. Similar comments for several other figures as well including bars in Figure 3A, lines in Figure 1A, bars in Figure 2C, etc.

Reviewer #3:

Remarks to the Author:

Synthetic energy sensor AMPfret deciphers adenylate-dependent AMPK activation mechanism.

The authors have harnessed FRET technology to develop a new AMPK construct that reliably reports the activation state of AMPK continuously in vitro and in cellulo. This technique could be utilized for high-throughput drug screening and cellular spatiotemporal analysis of AMPK activation state and location during the cell's life cycle.

Previous structural studies have mainly focused on the active form of AMPK; there is a major gap in our understanding of the ATP-bound or inactive form of AMPK. In this study, the authors build on their previous SAXS data to detect a major structural rearrangement of AMPK upon AMP/ADP binding using the newly developed AMPfret. This technique allows the authors a unique ability to determine binding constants for AMP, ADP and ATP without competing nucleotides which are essential in normal kinetic experiments. The AMPfret technique may be a useful tool for future AMPK studies, however as it only uses 1 of the possible 12 AMPK heterotrimer combinations its utility for detailed mechanistic understanding of AMPK both in vitro and in the cell is somewhat restricted.

Specific points

1 Why was rat $\alpha 2$ and $\gamma 1$ used instead of all human or all rat subunits? Will this impact how the AMPfret construct interacts in a cellular context?

2 A major weakness of AMPfret is that it utilizes only $\alpha 2\beta 2\gamma 1$ containing heterotrimers. Have the authors tried to adapt AMPfret for $\alpha 1$, $\beta 1$ or any other γ -subunits? If so, why have they failed?

3 Page 3 Line 57: CamKK2; capitalize the M – CaMKK2. Please change throughout the text line 57 the authors need to cite Hurley et al 2005 J Biol Chem. 280 (32): p. 29060-6. Line 60 It is not entirely correct to refer to individual cystathionine beta synthase (CBS) as domains rather pairs of CBS sequences form stable Bateman domains. Line 61 The work of Chen et al. probably indicates CBS4 AMP is exchangeable. To describe CBS2 as inactive is a bit harsh on its function. While it does not bind nucleotide due to the absence of a critical Asp it nevertheless contributes basic residues that recognize the AMP phosphates and are important for transduction.

4 Page 4 line 75 delete "first". Line 80 reference 29 should be dropped as it is not on Pubmed.

5 Page 5 There are many curious points to mention in Figure 1c. 1. It is unclear why the authors have used the structural models 4EAI and 4EAK when they show no change at the C-terminus of both the β and γ -subunits (where the FPs are located) when bound to AMP or ATP. These

structures do not support a large structural rearrangement between the two nucleotide bound forms. 2. These structures have been integrated into 5ISO an $\alpha 2\beta 1\gamma 1$ model, when the relevant $\alpha 2\beta 2\gamma 1$ model could have been used (6B2E). 3. The labelling of the FPs doesn't match the figure legend.

6 Page 7 line 153 write "conformational changes are independent of the activation loop phosphorylation status". Line 167 The authors suggest synergistic activation of AMPK when incubated with A-769662 and AMP (Figure 2e), which has been shown for $\alpha 1\beta 1\gamma 1$ by Scott (ref 15). However, it has been reported that these two compounds are unable to work synergistically with $\alpha 2\beta 2\gamma 1$ AMPK (reference 58 and Ngoei et al., 2018. Cell Chem. Biol.

DOI:10.1016/j.chembiol.2018.03.008), but instead activate in an additive manner. The data in Figure 2e show only a very small increase with A-769662 (although dose-dependent) and that AMP is the main contributor to the signal – in previous studies mentioned, synergistic activity has been more than 500-fold higher than basal activity and 10-fold higher than either compound alone. It indicates that the activation is additive. It is unclear if the AMPfret was unphosphorylated for this experiment, a requirement for synergistic activation of AMPK. An accompanying ACC blot would be useful to see full synergistic output. In Figure 2d it is puzzling why the 991 response is so different between the AMPfret constructs. Why does the position of the YFP make such a difference? Line 172 At face value the result showing a lack of FRET with NADH is welcome, but one is left with the lingering doubt that such a negative result may have been a function of the AMPfret construct.

7 Page 8 first paragraph. The results seem to show that the AMPfret biosensor could be used to screen for $\beta 2$ containing isoform activators acting at the ADAM site. However, the authors have only demonstrated this for a single example. Line 186 The mutations made to disrupt the CBS sites should be mentioned in the text. The observation that disrupting site 1 and 4 had differential effects on AMP and ADP sensitivity is particularly interesting and potentially a major insight but it may be influenced by the disrupting mutations used. It would be valuable to mutate the conserved Asp residues to see if this conclusion holds up with a more conserved D to A mutation? Line 187: Our experiments clearly identify CBS3 as the main site of..... This is not a new finding, see references. Oakhill et al., 2010. PNAS. DOI:10.1073/pnas.1009705107; Chen et al., 2012. NSMB. DOI:10.1038/nsmb .2319. Line 190 mutating CBS4, which abolished the FRET response triggered by ADP, but not by AMP is an important observation. It would seem consistent with nucleotide exchange at CBS4

8 Page 9 There is 2.0 mM MgCl₂ in the AMPfret purification buffers, was this removed before measuring the effect Mg²⁺ has on adenylate binding? This result indicates that if Mg²⁺ is added in excess in kinetic assays, then it wouldn't interfere with the other nucleotides (AMP or ADP) in the experiment. Thus, validating previous kinetic studies where Mg²⁺ is added in excess. Can the authors comment on this?

9 Page 10. Line 187: Our experiments clearly identify CBS3 as the main site of..... This is not a new finding, please change sentence accordingly and add appropriate references. Oakhill et al., 2010. PNAS. DOI:10.1073/pnas.1009705107; Chen et al., 2012. NSMB. DOI:10.1038/nsmb .2319 The AMPfret construct provides an opportunity to test the relative contributions of AMP and ADP to the conformational change in the kinase. They could potentially answer the question where on the one hand Hardie's laboratory (ref 13) have argued AMP is the true activator whereas Oakhill and Carling have provided evidence for ADP being the major contributor.

10 Page 11. Is cellular resolution high enough to visualize specific localization of AMPK; for instance, AMPK accumulation on the lysosome as suggested in Zhang et al., 2017. Nature. 548 (112-116) DOI:10.1038/nature23275? If AMPfret can't localize to the nucleus, doesn't this invalidate AMPfret for localisation studies? i.e. AMPfret has altered protein-protein interactions blocking AMPK's normal activity in the nuclei, therefore other important protein-protein interaction might be blocked as well. Can the authors comment on this?

11 Page 12 Figure 5. In using the AMPfret construct in cells it would be important to know whether the construct itself disturbed the cell. What happens if a kinase dead version is used? It is not clear whether the AMPfret construct in cells works for drugs. Figure 5 the ACC loading blots have not come from the same gel as the pACC blots (band details do not match).

12 Page 14 Line 320: AMPfret retains wild-type..... Authors should add here that in cells AMPfret doesn't translocate to the nucleus like wild-type.

Overall this is a well written manuscript and the authors have provided strong support for their conclusions. However, the areas where their findings potentially contribute to a new understanding of AMPK are insufficiently developed.

1 They need to demonstrate the differential regulation by AMP and ADP at the CBS sites they have identified is not restricted to the mutant constructs they have used.

2 They need to answer the question of the relative contributions of AMP and ADP in modulating the conformation of AMPK

3 The general utility of the AMPfret construct is restricted to $\alpha 2\beta 2\gamma 1$ containing heterotrimers. This is a serious limitation for drug screening.

Reviewer #4:

Remarks to the Author:

The authors have developed an interesting and original tool, AMPfret, for the direct study of AMPK conformational changes upon binding of nucleotides. They extensively characterize and validate AMPfret in vitro. Furthermore, they use AMPfret for a set of in vitro experiments that have an important impact on the understanding of the biochemical and physiological function of AMPK. An important advantage of genetically codified FRET sensors, as AMPfret, is the capacity of performing experiment in living cells. The set of AMPfret experiments in living cells here presented is limited in its scope.

Please find here below my specific comments:

1. Statistical analysis. The authors report in the Materials and Methods sections (lines 625 and 626) that 'where not stated otherwise, statistical significance was calculated by a two-tailed Student's t-test'. Requirements for the applicability of the t-test are that data are normally distributed and that are sampled from populations with equal variances. The authors do not report the results for tests of normality of distribution and homoscedasticity. Furthermore, in several instances multiple comparisons are performed (for example Fig.2, panel E; Fig.3, panel A; Fig.4, panels A and B; and Fig. 5, panel F). Multiple comparisons require an appropriate correction such as Bonferroni. If data are normally distributed, ANOVA should be first taken in consideration, followed by appropriate post hoc tests.

2. Lines 197 and 198. In order to assess proper agreement between the results obtained with AMPK activity assays of the CBS mutants and AMP-induced FRET changes, FRET experiments should be conducted in presence of all the concentrations of AMP and ADP used in the AMPK activity assays. Currently, FRET results, as reported in Fig.3 panel A, are only for 30microM AMP and 200microM ADP.

3. Line 228. The authors write that they have used AMPfret 2.1 for the experiments reported in Fig.4, panel B. The figure legend (line 900) reports AMPfret 2.0. Also, Is AMPfret 1.0 been used for obtaining the results shown in Fig.4, panel A? This has not been specified either in the text or in the figure legend.

4. Experiment in living cells. The authors report in the Materials and Methods section (lines 603 to 613) that an incubation chamber has been used for the confocal FRET experiments. Please specify the temperature at which the confocal experiments have been conducted as well the composition of the medium. Further acquisition parameters are required such as power of the excitation laser line, image pixel resolution, scanning frequency and temporal image resolution. The authors write that the signal of CFP and YFP fluorescences have been extracted and the FRET signal per single cells calculated. How? What is the FRET ratio plotted in Figure 5, panels C and E? Please, define it.

5. Please add dimension bars in Fig.5 panels A and D.

6. Lines 259, 260 and Fig.5. There is minimal decrease of the fluorescence intensity of the donor upon AICAR exposure (Fig.5, panel B). This is concerning. Upon FRET, increase of the sensitised emission of the acceptor and decrease emission of the donor are expected. Can the authors provide the average values of the fluorescence intensities in the donor channel (478+/-10nm) and acceptor channel (530+/-10nm) before and upon exposure to AICAR for the experiments plotted in panel C and E?

7. Are the n values in the legend of Fig.5, panel C and E referring to the number of single analysed cells? If so, how many independent experiments have been performed for the data shown in panel C and E?
8. The behaviour of AMPfret 2.1 in living cells has been assessed by exposure to the AMPK activator AICAR. This is interesting but it does not allow comparison to the in vitro behaviour since 991 was used instead. An experiment in living cells with exposure to 991 should be conducted. Furthermore, the capacity of AMPfret 2.0 of reporting changes of AMPK activity should also be assessed by inhibition of mitochondria respiration.

Synthetic Energy Sensor AMPfret Deciphers Adenylate-dependent AMPK Activation Mechanism

Martin Pelosse, Cécile Cottet-Rousselle, Cécile Bidan, Aurélie Dupont, Kapil Gupta, Imre Berger and Uwe Schlattner

Point-by-point response to Reviewer's comments.

Reviewer #1 (Remarks to the Author):

The authors have made use of the conformational change that occurs on binding of AMP or ADP to AMPK (first detected by their group by SAXS in 2008) to develop a FRET sensor based on the alpha-2/beta-2/gamma-1 complex of AMPK. This allows the conformational change that occurs upon binding of AMP or ADP to be detected and monitored, independently of changes in phosphorylation of Thr172 and hence overall kinase activity. Most of the manuscript describes the use of these sensors in cell-free assays, and this has generated some useful information such as: (i) confirmation of previous proposals that only free ATP, rather than the MgATP complex, competes with AMP or ADP for binding to the gamma subunit; (ii) confirmation that previously reported effects of NAD can be accounted for by contamination with AMP; (iii) analysis of the contributions of the individual gamma subunit sites to the binding of AMP and ADP [although see point (4) below]. However, although the FRET sensor is also tested in intact cells, the statement that the authors themselves make on lines 441-442 ("The resulting 20% range in the AMPfret 2.1 FRET signal *in vitro* may still be limiting for certain applications in living cells") seems to indicate that the relevance of the current generation of sensors to intact cell studies may be somewhat limited. Nevertheless, this manuscript does describe a potentially useful addition to the armory of experimental tools used to study the AMPK system and, with attention to the points raised below, would be of interest to the AMPK aficionados.

We thank the Reviewer for this positive evaluation of our work and for the insightful comments. Indeed, in this manuscript, we utilize our novel AMPfret sensor to provide unique and novel mechanistic insights of allosteric AMPK activation *in vitro* in cell-free assays. At the same time, we also provide here compelling data underscoring the utility of our AMPfret *in vivo* when deployed in cellular assays.

MAJOR POINTS:

1. Fig. 2a, and elsewhere: the authors should consider the use of semilog plots (i.e. \log_{10} [AMP] versus FRET ratio), which are often better at displaying differences in responses, such as the rightward shift when the output was measured using p-ACC:ACC in the presence of ATP.

We agree with the Reviewer that the use of semi-log plots could indeed have been an alternative when comparing two traces as it is the case in Fig. 2a. We have scrutinized this alternative but found that the linear scale is superior in reporting the FRET and AMPK activity changes (and their error margins) within the micro-molar range increase (0-20 μ M) in AMP concentration that we analyze. Thus, we prefer to maintain the original representation.

2. Line 236: can the authors state why they think that 1 μ M AMP, 100 μ M ADP and 4 mM ATP approximate to physiological conditions?

Exact concentrations of free adenylates will depend on the cell type and the physiological conditions, as well as the quantity of structurally bound adenylates. However, the consensus is that under non-stressed, basal conditions, $[ATP]_{free}$ is about 1-2 orders of magnitude above $[ADP]_{free}$, and that this is about 2 orders of magnitude above $[AMP]$. This is consistent with measurements in cells and tissues (e.g. extended refs. 21, 66, 67 and 68), including our new adenylate measurements in HEK293 cells (Fig. 5d). Moreover, it is also consistent with calculations using the catalytic parameters of relevant enzymes, i.e. assuming equilibrium reactions of creatine and adenylate kinases (cf. model calculations for pool sizes of adenylates with decreasing “high-energy” phosphate pools, reproduced below; Fig. 3.2, p. 309 in Neumann et al 2007, Signaling by AMP-activated protein kinase. In: Molecular System Bioenergetics (V. Saks, ed.), Wiley-VCH, Weinheim, Germany).

We have considered ATP concentration in healthy cells with high energy turnover (myocytes, neurons etc.), which is about 4-6 mM (less in more quiescent cells) and have then chosen ADP and AMP concentrations accordingly. Thus, experiments in Fig. 4b (right part) are physiological because they follow these considerations.

3. Line 263: can the authors state what they mean by the nuclear localization signal in alpha-2? The authors should discuss Kazgan et al (2010) Mol Biol Cell 21:3433, which defined a nuclear export sequence (NES) at the C-terminus of alpha-2 that is also conserved in alpha-1. How has the NES in alpha-2 been affected by the construction of their FRET sensors, especially the second generation?

Very little is known about the mechanisms leading to nuclear localization of both alpha1 and alpha2 AMPK complexes (which may be regulated differentially). In our present manuscript, we speculate that recognition of a putative nuclear localization sequence (NLS) maybe hampered by our bulky fluorescent tags, which could be remedied by engineering into our sensor a bona fide NLS.

We thank this reviewer for drawing our attention to the Kazgan et al. (2010) paper (now introduced as ref. 57), which is in fact the only study directly addressing nuclear AMPK localization. It identifies a nuclear export signal (NES) in the alpha C-termini, the modification of which leads to nuclear accumulation of AMPK. We posit that this may be another option to increase AMPfret sensor levels in the nucleus. In our AMPfret sensor constructs, the described NES is retained (we had removed merely two residues, Ala551 and Arg552, in this context not relevant, at the extremity of the alpha2 C-terminus). Of note, we have re-examined nuclear fluorescence in a new series of experiments with HEK293 cells (see Figs 5a-d). We observe nuclear YFP fluorescence clearly above background levels, albeit too low to reliably determine FRET (included as Suppl. Fig. 7).

Thus, nuclear import of AMPfret may just be rather ineffective. We have added the following text comprehensibly commenting on this matter in our revised manuscript:

Lines 268ff:

Nuclear AMPfret fluorescence was present, but not sufficient to reliably calculate FRET (Suppl. Fig. 7). We speculate that a putative nuclear localization signal present in $\alpha 2$ may be obstructed by the FP in the tagged construct, while a nuclear export signal identified in the α -C-terminal part remains active³⁷.

Line 344:

Nuclear import of AMPfret is reduced, but still occurring.

4. Fig. 5: although it is reasonable that they should initially assess the efficacy of their sensor in intact cells using AICAR, the real challenge is whether it can detect changes in the ratios of endogenous nucleotides when compounds that disturb cellular energy balance are added. Why did they not attempt this?

Although ZMP and AMP generated from AICAR within a cell should work similar as AMP generated during energy stress, we agree with the Reviewer that direct proof with altered endogenous adenylates was lacking in the original version of our manuscript. In its revised version, we therefore now included new data from experiments with HEK293 cells. In these new experiments, in addition to using AICAR, we have inhibited glycolysis with 2-deoxyglucose (2-DG). 2-DG induces energy stress by decreasing ATP/ADP and ATP/AMP ratios (new Fig. 5d). Consistent with what would be expected, AICAR increases AMPfret signals faster than 2-DG, since its cellular products directly act on AMPK (new Fig.5c). However, with 2-DG, much higher AMPfret signals can be reached, since in this case not only does AMP increase (like ZMP with AICAR), but, at the same time, ATP decreases drastically. Moreover, while AMPfret signals with AICAR rapidly reach a plateau phase, possibly dependent on AICAR import and its conversion into ZMP and AMP, signals with 2-DG continue to increase over 1 hour, indicating steady degradation of ATP/ADP and ATP/AMP ratios. These novel data provide direct evidence that AMPfret can indeed function as readout of endogenous adenylate concentrations and cellular energy state. These experiments are detailed in different new paragraphs of our revised manuscript:

Methods section (lines 632ff):

HEK293t cells (ATCC) were cultured in DMEM (Gibco 41965039) containing 4,5 g/L glucose, 10% SVF, penicillin and streptomycin. For transfection of cells at ~80% confluence, medium was replaced by OptiMEM (Gibco) and MultiMam-derived AMPfret 2.1 coding plasmid was transfected using either Lipofectamine 2000 (Invitrogen) for 5 to 6 hours (3T3-L1 and HeLa cells) or PolyFect transfection reagent (Qiagen) following manufacturer recommendations (HEK293t cells). For extracts, cells were grown to ~80% confluence in Ø35mm dishes before replacing the medium by either fresh complete medium (control), complete medium with AICAR (1mM final), or 2-DG medium (containing 3.3g/L 2-DG, 1 g/L glucose instead of 4,5 g/L glucose), and cells grown for variable time until lysis. For AMPfret analysis, cells cultured in 8-well Labtek (Nunc) in complete medium were observed by confocal microscopy. Medium was then replaced by complete medium containing AICAR (1 mM final) or 2-DG (3.3g/L 2-DG and 1g/L glucose) without moving the 8-well plates.

Results section (lines 285ff):

To further explore whether AMPfret can be used for detection of both allosteric AMPK activation and spatiotemporal analysis of endogenous adenylate ratios, we inhibited

glycolysis in HEK293t cells (Fig. 5c) by using 2-deoxyglucose (2-DG, 3.3 g/L) at low glucose (1 g/L) as compared to normal glucose (4.5 g/L). Again, AMPfret responded by an increase in the FRET signal, but much slower as compared to AICAR, with a half-maximal response occurring only after ~20 min and reaching saturation at times >50 min (not shown). However, the maximal FRET signal achieved was about twice as high as with AICAR. These data are consistent with ongoing energy depletion induced by 2-DG, slowly decreasing ATP and increasing ADP and AMP levels, which then translate into allosteric AMPK activation. Indeed, direct quantification of adenylates by HPLC confirmed that the AMPfret signal 50 min after 2-DG addition corresponds to a 2- and 10-fold increase of ADP/ATP and AMP/ATP ratios, respectively (Fig. 5d). Different to 2-DG, AICAR uptake directly generates the allosteric activators ZMP and AMP, which probably causes the more rapid activation, but does not alter ATP and ADP levels. This may explain the lower maximal AMPfret signal. These data provide compelling evidence that AMPfret indeed affords readout of endogenous adenylate ratios.

Discussion section (line 456ff):

The different time course and magnitude of the AMPfret response to AICAR and 2-DG illustrate how AMPfret reports changes in endogenous adenylate ratios and thus cellular energy state.

5. Discussion section: the previous paper that is perhaps most relevant to the discussion of their results is Gu et al (2017). Although it used different experimental approaches to address the roles of the three CBS sites, some of the mutations used to eliminate binding to the individual sites even appear to have been the same as those used in this paper. It would be helpful for the Discussion section of this manuscript to include a more direct comparison of the results of Gu et al (2017) with those of the present authors, as well as their interpretation (which in some cases appears to be conflicting).

We agree with the Reviewer that results from Gu et al. (ref. 24) from the Melchers group which describe adenylate sensing at the CBS sites, are highly relevant for part of our study. We cite them already at multiple instances (see lines indicated below). To delete AMP binding sites at the CBS sites, both studies apply those mutations from our earlier collaborative work with the Wu group that showed the strongest effect on AMP-stimulated AMPK activity (Fig.3 in Chen et al. 2012 Nat Struct Mol Biol, ref. 15). While mutations at CBS1 (L128D/V129) and CBS3 (V275G/L276) are identical in our studies, we used S315P for CBS4 instead of I311D chosen by Gu et al. Importantly, they use a more indirect adenylate competition binding assay (citation line 327) that works only for AMP, while we can directly monitor conformational changes by using AMP, ADP, and ATP. The trend pointed to by most findings in the Gu et al. paper is fully consistent with our present data:

- Both studies report AMP affinities in the very low micromolar range and suggest CBS3 being most relevant for AMP-dependent activation (high affinity site) as compared to CBS1 (low affinity) (citation lines 76, 394), although AMP affinities in some experiments by Gu et al are very low (citation line 378)
- Both studies report differences between CBS binding of free ATP and MgATP (citation line 224, 381)
- Consistent with our suggestion based on AMPfret data, HDX-MS experiments by Gu et al. indicate flexibility at or close to the beta and gamma C-termini (citation line 334), and a coupling between the gamma-subunit CBS domains and the beta-subunit ADaM site, supporting a similar activating conformation induced by both CBS and ADaM site ligands

(comment added in line 434, comment added):

This fact, together with the almost identical outcome of our construct screening with AMP and 991, may reflect overall similar intramolecular and intermolecular rearrangements within the AMPK heterotrimer, irrespective of the initial activator or binding site involved, as also suggested earlier²⁴.

However, marked differences exist between the two studies:

- Mutation of CBS4 by Gu et al. inhibits AMP-dependent activation at the exchangeable CBS3 site (their Fig. 5B), while we observe such a cross talk only for ADP-dependent activation (line 406). Possibly, this is a consequence of the different CBS4 mutations used (I311D instead of S315P). Although both eliminate AMP-binding to CBS4 (ref. 15), the I311D mutation may directly affect the neighboring CBS3 site.
- Gu et al. did not observe by HDX-MS differences in AMPK conformation when incubating with adenylate mixtures representing “non-stressed” and “stressed” states. In contrast, we report here clear changes in AMPfret signal under such conditions. We posit that this originates from us using a different, truly “unstressed” adenylate mixture with lower ADP and AMP concentrations as compared to the Cu et al study (new citation line 243, comment added):

Next, approximating physiological conditions, we choose a baseline resting state of 1 μ M AMP, 100 μ M ADP and 4 mM ATP (Fig. 4b), i.e. lower AMP and ADP concentrations as used earlier for “unstressed” conditions²⁴.

MINOR POINTS:

6. Line 60: whether AMP exchanges with ATP at two of the four CBS sites remains the subject of debate. Gu et al (2017) proposed instead that the CBS1 site may normally always have ATP bound. It might be better to state: “AMP and ADP levels increase and competitively replace ATP at up to two of the four cystathionine beta synthase (CBS) domains”.

We agree with the Reviewer and have included the suggested change in our revision (line 59).

7. Line 62: It might be more accurate to describe the CBS2 site as “unoccupied” rather than “inactive”.

We agree with the Reviewer and have included the suggested change in our revision (line 63).

8. Line 93: what exactly do they mean by “common FRET sensors”? Perhaps “existing FRET sensors” is a better description?

We agree with the Reviewer and have included the suggested change (to “existing”) in our revision (line 94).

9. Line 98: they had already described in the Introduction how the conformational change that occurs upon AMP binding to AMPK was discovered and defined, and I would suggest that the words “we had discovered” are superfluous here and should be deleted.

We agree with the Reviewer and have deleted this part of the sentence in our revision.

10. Line 187: although it was possible to identify which residues in gamma-1 were mutated in Supplementary Fig. 6a-c, to find out what they were replaced with it was

necessary to dig down into the Methods section. The authors should please provide details of the mutations that were made in a much more accessible fashion.

We apologize for our oversight. In our revised manuscript, we have properly identified the mutations in addition to the Methods section (line 526) also in the Results section (line 193) and Suppl. Fig. 6a-c.

Reviewer #2 (Remarks to the Author):

The article by Pelosse et al describes an interesting and clever tool where the authors use a genetically encoded FRET system to report on ligand-induced conformational changes in the multisubunit energy sensing kinase AMPK. Based on many prior studies, AMPK is known to be a dynamic and flexible complex and the authors use their system, AMPfret, to monitor FRET changes in response to binding of known ligands. Of particular interest is the ability to use AMPfret in live cells to get a kinetic read of AMPK engagement, and the ability to carefully titrate nucleotide ratios in vitro (including +/- Mg) to interrogate AMPK response.

Major points:

1) Although this paper provides many interesting insights, a number of questions remain regarding interpretation and utility of this AMPfret system. First, it is interesting to note that AMPfret yields an increase in FRET signal upon both nucleotide binding to the gamma subunit, and allosteric activator binding to the distal ADaM site. The authors suggest that this may reflect similar rearrangements in AMPK, irrespective of the initial activator or binding site involved – and this observation leads to the proposal that AMPfret can be used as a novel screen for AMPK activators. It would be interesting and important to demonstrate if a similar FRET change occurs upon inhibitor binding to the ATP site. A potent and available tool compound known to inhibit AMPK is staurosporine. If staurosporine binding also leads to an increase in FRET, then this may require that changes be made to the discussion regarding interpretation of the nature of the FRET signal – and may demonstrate that this system reports generally on ligand binding rather than reporting on an ‘active’ conformation.

We agree with the Reviewer that also binding in the active site pocket, e.g. by a kinase inhibitor like staurosporin, will induce a conformational change. As suggested by the Reviewer, we have carried out new experiments with the kinase inhibitor staurosporine, which forces the alpha kinase domain in an active conformation. However, in our experiments, we did not observe an increase in the AMPfret signal, rather, there appeared to be a slight signal decrease (Fig. 2d). The induced conformational change appears to be local, likely restricted to the kinase domain and is not transmitted to other subunits. Thus, our sensor is exclusively sensitive to active conformations induced in the entire heterotrimeric complex by the binding of allosteric activators. We have included this additional information in the revised version of our manuscript:

Line 172ff:

Also, local conformational changes induced in the α -subunit kinase domain by the kinase inhibitor staurosporine did not translate into an activating conformation of the entire heterotrimeric complex as detected by AMPfret (Fig. 2d).

Line 434ff:

... These are different from conformational changes triggered by binding at the kinase domain active site, as staurosporine does not induce an AMPfret signal.

2) In results, paragraph 2, line 119, the authors describe a construct, AMPfret 2.0, that was biochemically characterized to confirm it retained endogenous affinity for AMP – yet it appears that this confirmation was made using FRET. It would be better to use an independent technique – aside from the FRET system being developed in this paper – to validate the protein FRET construct. An independent binding assay would be useful to build confidence that the FRET signal of these constructs correlates with affinity.

We have carried out extensive characterization of AMPfret 1.0, correlating the FRET signal with AMPK activation by radioactive assay and immunodetection of P-ACC (see Figs. 2a, 3, Suppl. Fig. 6). For AMPfret 2.0/2.1, we present end-point data for 20 μ M AMP (see Fig. 1a,b; Suppl. Fig. 3a,b). In our revised manuscript, we clarify this as follows:

Line 120ff:

AMPfret constructs retained endogenous AMPK affinities for AMP and ADP, as shown extensively for AMPfret 1.0 (Suppl. Figs 4, 6).

3) Discussion in the manuscript suggests that AMPfret can be used as a cellular sensor of energetics, and data from Figure 5 shows an exciting kinetic trace for cells treated with AICAR. My question and concern is regarding discussion of how this can be used. Although treatment with AICAR shows a 7% increase in signal for AMPfret, I don't know how to interpret this in terms of cellular energetic state. Can this truly be used to assess the concentrations of different adenylates as suggested? What would these values be for the experiment shown? If the authors truly propose that this probe can be used to monitor cellular energy state (line 299), then it would be nice to see at least some rough quantitation from in-cell studies (in addition to the normalized kinetics shown).

This remark addresses the issue of relative and absolute differences. FRET sensors like AMPfret report relative changes between two different conditions and provide temporal and spatial information at a single cell level. In our case AMPfret reports on increased or decreased ATP/ADP and ATP/AMP ratios. Absolute values can only be obtained e.g. for adenylates by extraction/HPLC analysis or NMR spectroscopy but will lack single cell spatiotemporal information. One may try to somehow “calibrate” FRET signals, but this will remain a crude estimate and is rarely done in the existing literature. Thus, both approaches provide valuable information on the cellular energy state and are complementary. To provide an estimate, we have now carried out new experiments with HEK293 cells and include this novel data in the revised version of our manuscript (novel Figs 5a-d). In addition to using AICAR, we have now inhibited glycolysis with 2-deoxyglucose (2-DG). 2-DG induces energy stress by decreasing ATP/ADP and ATP/AMP ratios, which we have quantified (new Fig.5d). Consistent with what would be expected, AICAR increases AMPfret signals faster than 2-DG, since its cellular products directly act on AMPK (new Fig. 5c). However, with 2-DG much higher AMPfret signals can be reached, since here not only AMP increases (like ZMP with AICAR), but also ATP decreases drastically. Interestingly, while AMPfret signals with AICAR rapidly reach a plateau phase, possibly dependent on AICAR import and ZMP/AMP degradation, signals with 2-DG continue to increase over 1 hour, indicating steady degradation of ATP/ADP and ATP/AMP ratios. These new data provide compelling direct evidence that AMPfret can indeed sense and report faithfully on

endogenous adenylate concentrations and cellular energy state. These novel experiment is detailed in different new paragraphs in our revised manuscript:

Methods section (lines 632ff):

HEK293t cells (ATCC) were cultured in DMEM (Gibco 41965039) containing 4,5 g/L glucose, 10% SVF, penicillin and streptomycin. For transfection of cells at ~80% confluence, medium was replaced by OptiMEM (Gibco) and MultiMam-derived AMPfret 2.1 coding plasmid was transfected using either Lipofectamine 2000 (Invitrogen) for 5 to 6 hours (3T3-L1 and HeLa cells) or PolyFect transfection reagent (Qiagen) following manufacturer recommendations (HEK293t cells). For extracts, cells were grown to ~80% confluence in Ø35mm dishes before replacing the medium by either fresh complete medium (control), complete medium with AICAR (1mM final), or 2-DG medium (containing 3.3g/L 2-DG, 1 g/L glucose instead of 4,5 g/L glucose), and cells grown for variable time until lysis. For AMPfret analysis, cells cultured in 8-well Labtek (Nunc) in complete medium were observed by confocal microscopy. Medium was then replaced by complete medium containing AICAR (1 mM final) or 2-DG (3.3g/L 2-DG and 1g/L glucose) without moving the 8-well plates.

Results section (lines 285ff):

To further explore whether AMPfret can be used for detection of both allosteric AMPK activation and spatiotemporal analysis of endogenous adenylate ratios, we inhibited glycolysis in HEK293t cells (Fig. 5c) by using 2-deoxyglucose (2-DG, 3.3 g/L) at low glucose (1 g/L) as compared to normal glucose (4.5 g/L). Again, AMPfret responded by an increase in the FRET signal, but much slower as compared to AICAR, with a half-maximal response occurring only after ~20 min and reaching saturation at times >50 min (not shown). However, the maximal FRET signal achieved was about twice as high as with AICAR. These data are consistent with ongoing energy depletion induced by 2-DG, slowly decreasing ATP and increasing ADP and AMP levels, which then translate into allosteric AMPK activation. Indeed, direct quantification of adenylates by HPLC confirmed that the AMPfret signal 50 min after 2-DG addition corresponds to a 2- and 10-fold increase of ADP/ATP and AMP/ATP ratios, respectively (Fig. 5d). Different to 2-DG, AICAR uptake directly generates the allosteric activators ZMP and AMP, which probably causes the more rapid activation, but does not alter ATP and ADP levels. This may explain the lower maximal AMPfret signal. These data provide compelling evidence that AMPfret indeed affords readout of endogenous adenylate ratios.

Discussion section (line 456ff):

The different time course and magnitude of the AMPfret response to AICAR and 2-DG illustrate how AMPfret reports changes in endogenous adenylate ratios and thus cellular energy state.

4) Regarding CBS mutants, unless I misinterpreted Supplementary Figure 1 and 6, CBS3 mutant appears significantly functionally impaired both in the absence and presence of AMP. This seems counter to the text in lines 186 and 187 and makes interpretation of data concerning this mutant uncertain.

These very same mutations have been used in earlier publications to specifically delete the binding of allosteric activators to CBS3 (refs 15, 24). These studies also show that such mutations abolish AMPK activation by AMP without affecting AMPK catalytic activity in absence of AMP, a fact also observed earlier (Scott et al. 2004 J Clin invest 113:274). Also our CBS3 AMPfret mutant is not “kinase dead”, since it can still be activated by T172

phosphorylation with CaMKKbeta to phosphorylate ACC in absence of AMP, albeit of course at much lower levels (Suppl Figs 1e and 6f,g,h). To clarify, we have added in Methods:

Line 525:

Mutations at the $\alpha 2T172$ (T172D and T172A) and at three $\gamma 1$ CBS sites shown to abrogate adenylate binding^{17, 24} (CBS1: L128D + V129D, CBS3: V275G + L276G and CBS4: S315P),

...

5) Given the panel of AMPfret constructs described, it would have been interesting to see if some of the constructs that were non-FRET-responsive to AMP were instead responsive to 991 or staurosporine. This could reveal selective sensors for different binding sites and would increase the potential breadth, utility, and impact of these AMPfret sensors.

We thank the Reviewer for this very valuable comment. The Reviewer's suggestion could have led to the identification of specific AMPfret constructs for the exclusive screening of interactors at either ADaM or CBS sites. Indeed, we have done such a screen for 991, which is now included as novel Suppl. Table 1 in the revised version of our manuscript. Interestingly, the very same constructs sensitive to AMP, represented by AMPfret 1.0 and 2.0, were also the most sensitive to 991, except that in case of 991 AMPfret 2.0 was much more sensitive than AMPfret 1.0, This is probably due to the fact that the ADaM site is situated between the alpha- and beta-subunits, both tagged in the AMPfret 2.0 series. We introduce this issue in the revised manuscript in the following section:

Results line 164ff:

A re-screening of our initial constructs for their response to 991 confirmed that those most responsive to AMP are also most responsive to 991 (Suppl. Table 1), suggesting that both allosteric activators trigger a similar activating AMPK conformation.

Discussion line 431ff:

This fact, together with the almost identical outcome of our construct screening with AMP and 991, may reflect overall similar intramolecular and intermolecular rearrangements within the AMPK heterotrimer, irrespective of the initial activator or binding site involved, as also suggested earlier²⁴.

Minor comment:

1) In Supplementary Figure 6E, it would be helpful to directly label the blue and orange lines as +/- AMP in the actual figure rather than simply the figure legend. Similar comments for several other figures as well including bars in Figure 3A, lines in Figure 1A, bars in Figure 2C, etc.

We thank the Reviewer for this helpful suggestion and explain now all color labels alongside the Figures in the revised version of our manuscript.

Reviewer #3 (Remarks to the Author):

The authors have harnessed FRET technology to develop a new AMPK construct that reliably reports the activation state of AMPK continuously *in vitro* and *in cellulo*. This technique could be utilized for high-throughput drug screening and cellular spatiotemporal analysis of AMPK activation state and location during the cell's life cycle. Previous structural studies have mainly focused on the active form of AMPK; there is a major gap in our understanding of the ATP-bound or inactive form of AMPK. In this study, the authors build on their previous SAXS data to detect a major structural rearrangement of AMPK upon AMP/ADP binding using the newly developed AMPfret. This technique allows the authors a unique ability to determine binding constants for AMP, ADP and ATP without competing nucleotides which are essential in normal kinetic experiments. The AMPfret technique may be a useful tool for future AMPK studies, however as it only uses 1 of the possible 12 AMPK heterotrimer combinations its utility for detailed mechanistic understanding of AMPK both *in vitro* and in the cell is somewhat restricted.

We thank the Reviewer for his positive evaluation of our study. There are indeed very few data on the inactive (ATP-bound) AMPK conformation and its switch to an active conformation. Our data suggest that there is a large similarity between active conformations triggered by allosteric activators, irrespective whether CBS or ADaM sites are involved.

Specific points

1. Why was rat $\alpha 2$ and $\gamma 1$ used instead of all human or all rat subunits? Will this impact how the AMPfret construct interacts in a cellular context?

We agree with the Reviewer that in future studies, only sequences from the same species should be used. We maintained such chimeric constructs for historic reasons and for consistence, as they had been used in several previous studies (ref. 48: Suter et al. 2006 J. Biol. Chem. 281; ref. 19: Riek et al. 2008 J. Biol. Chem. 283, 18331; ref. 15: Chen et al. 2012 Nat Struct Mol Biol 19). Given the very high degree of sequence identity between human and rat isoforms, this should not affect any crucial parameters like substrate or ligand affinities.

2. A major weakness of AMPfret is that it utilizes only $\alpha 2\beta 2\gamma 1$ containing heterotrimers. Have the authors tried to adapt AMPfret for $\alpha 1$, $\beta 1$ or any other γ -subunits? If so, why have they failed?

The Reviewer raises an important point which motivates us for future studies. Indeed, we are aware that studying differences between various isoform combinations is important for further mechanistic analysis. Since such isoform-dependent differences seem to exist for allosteric activators at the ADaM and potentially also at the CBS sites, our sensor technology would be in fact ideally suited for such studies, and experiments are underway or will be performed in our laboratories in due course. There is no specific issue that would prevent such studies, since fluorescent tags are situated at the C-termini which in contrast to N-termini remain globally unchanged between isoforms (e.g. in the longer gamma2 and 3 isoforms as compared to gamma1). Furthermore, the modular design of our ACEMBL cloning/recombination system allows rapid generation of the necessary vectors. In the foundational, already data-rich study presented here, we chose one AMPK isoform combination for a comprehensive characterization from molecular *in vitro* all the way to cellular *in vivo* studies, revealing

unique new insights into AMPK mechanism in the process, firmly setting the stage for experiments studying isoforms in the future.

3. Page 3 Line 57: CamKK2; capitalize the M – CaMKK2. Please change throughout the text line 57 the authors need to cite Hurley et al 2005 J Biol Chem. 280 (32): p. 29060-6. Line 60 It is not entirely correct to refer to individual cystathionine beta synthase (CBS) as domains rather pairs of CBS sequences form stable Bateman domains. Line 61 The work of Chen et al. probably indicates CBS4 AMP is exchangeable. To describe CBS2 as inactive is a bit harsh on its function. While it does not bind nucleotide due to the absence of a critical Asp it nevertheless contributes basic residues that recognize the AMP phosphates and are important for transduction.

We thank the Reviewer for a careful reading of the manuscript. We have included all revisions as suggested.

4. Page 4 line 75 delete “first”. Line 80 reference 29 should be dropped as it is not on Pubmed.

We have deleted “first”. However, we are not sure whether ref. 29 (Hardie & Lin 2017 F1000Res 6) is the one the Reviewer intended to indicate. “F1000Research” articles and reviews are indexed by PubMed, PubMed Central, MEDLINE, Europe PMC, Scopus, Chemical Abstract Service, British Library, CrossRef, DOAJ and Embase.

5. Page 5 There are many curious points to mention in Figure 1c.

1. It is unclear why the authors have used the structural models 4EAI and 4EAK when they show no change at the C-terminus of both the β and γ -subunits (where the FPs are located) when bound to AMP or ATP. These structures do not support a large structural rearrangement between the two nucleotide bound forms.
2. These structures have been integrated into 5ISO an $\alpha 2\beta 1\gamma 1$ model, when the relevant $\alpha 2\beta 2\gamma 1$ model could have been used (6B2E).
3. The labelling of the FPs doesn't match the figure legend.

The Reviewer spots here an issue that we have also discussed among the authors when we prepared our manuscript, namely which published PDB structure would be appropriate to illustrate our sensor (Fig. 1c). Responding to the specific points raised by the Reviewer:

1. Our illustration is a schematic rather than a precise atomic model. We chose PDB ID 4EAI and 4EAK, because they are unique in representing the same AMPK species bound by either AMP or ATP, respectively. They do show differences in many parts of their structure. It is true that C-termini of these crystallized structures show few differences, but probably we detect already minor changes in the C-termini relative to each other and do not need a large structural rearrangement as suggested. In addition, these structures contain deletions directly at (γ) or close to (α , β) the C-termini, thus limiting conclusions on their respective orientation.
2. The $\alpha 2\beta 2\gamma 1$ structure (PDB 6B2E) was not available when we made this figure. Given the very high sequence homology between $\beta 1$ (in PDB 5ISO) and $\beta 2$ and the fact that the figure is a schematic used here to visualize the concept of FRET signal generation by the sensor, we prefer to keep the Figure as is.
3. We have corrected our error in the revised version of our manuscript.

6. Page 7 line 153 write “conformational changes are independent of the activation loop phosphorylation status”. Line 167 The authors suggest synergistic activation of AMPK

when incubated with A-769662 and AMP (Figure 2e), which has been shown for $\alpha 1\beta 1\gamma 1$ by Scott (ref 15). However, it has been reported that these two compounds are unable to work synergistically with $\alpha 2\beta 2\gamma 1$ AMPK (reference 58 and Ngoei et al., 2018. Cell Chem. Biol. DOI:10.1016/j.chembiol.2018.03.008), but instead activate in an additive manner. The data in Figure 2e show only a very small increase with A-769662 (although dose-dependent) and that AMP is the main contributor to the signal – in previous studies mentioned, synergistic activity has been more than 500-fold higher than basal activity and 10-fold higher than either compound alone. It indicates that the activation is additive. It is unclear if the AMPfret was unphosphorylated for this experiment, a requirement for synergistic activation of AMPK. An accompanying ACC blot would be useful to see full synergistic output. In Figure 2d it is puzzling why the 991 response is so different between the AMPfret constructs. Why does the position of the YFP make such a difference? Line 172 At face value the result showing a lack of FRET with NADH is welcome, but one is left with the lingering doubt that such a negative result may have been a function of the AMPfret construct.

We thank the Reviewer for these suggestions which we address as follows:

Figure 2e: We agree with the Reviewer that our wording is incorrect in this case. The data observed for a combination of AMP with A-769662 show a sort of additivity, and not true synergism. The small effect is probably due to the low affinity of beta2-containing AMPK for A-769662. We have corrected this in the revised version of our manuscript (see below). Further, our bacterially expressed AMPK is never phosphorylated at T172 (it may be partially phosphorylated at other residues); this would fit with additivity.

Line 167ff:

We further investigated the effect of A-769662 in presence or absence of AMP to account for potential cumulative effects (Fig. 2e). In the presence of 20 μ M AMP, A-769662 increased the FRET ratio in a concentration-dependent manner, supporting an additive effect of A-769662 and AMP for AMPK activation, consistent with previous reports^{43, 44}.

Line 354:

However, although α T172 phosphorylation is required for canonical AMPK activation, there is increasing evidence for an allostery-only activation by the sole additive or synergistic action of CBS and ADaM site activators^{17, 44, 63}.

Figure 2d: Globally, there are two reasons for the differences between the AMPfret constructs. First, AMPfret 2.1, as compared to 1.1, was engineered for superior sensitivity. Second, it contains fluorescent tags at the C-termini of alpha- and beta-subunits, which are precisely the subunits that form the ADaM site. We have added in the Supplementary Material of our revised manuscript a novel screen of different primary AMPK constructs for their reactivity to 991 (new Suppl. Table 1). This new data clearly shows that the 2.1 construct is by far most sensitive to 991.

7. Page 8 first paragraph. The results seem to show that the AMPfret biosensor could be used to screen for $\beta 2$ containing isoform activators acting at the ADaM site. However, the authors have only demonstrated this for a single example. Line 186 The mutations made to disrupt the CBS sites should be mentioned in the text. The observation that disrupting site 1 and 4 had differential effects on AMP and ADP sensitivity is particularly interesting and potentially a major insight but it may be influenced by the disrupting mutations used. It would be valuable to mutate the conserved Asp residues to see if this conclusion holds up with a more conserved D to A mutation? Line 187: Our

experiments clearly identify CBS3 as the main site of..... This is not a new finding, see references. Oakhill et al., 2010. PNAS. DOI:10.1073/pnas.1009705107; Chen et al., 2012. NSMB. DOI:10.1038/nsmb.2319. Line 190 mutating CBS4, which abolished the FRET response triggered by ADP, but not by AMP is an important observation. It would seem consistent with nucleotide exchange at CBS4

We have used four different AMPK activators that cover the different sites and mechanisms known, including CBS activators (AMP, AICAR), ADaM site activators binding to beta1 or 2 (991) or preferring beta1 (A-769662), as well as an indirect activator (metformin). All show the same behavior with our sensor exactly as predicted by the mechanistic models, and thus provide proof of principle for the utility of AMPfret as a screening tool. A more exhaustive testing of AMPfret for drug screening however is in our view beyond the scope of our present, already data rich study.

Mutations used to disrupt CBS sites have been used and validated earlier by different groups (ref. 15: Chen et al 2012 Nat Struct Mol Biol 19:716; ref. 24: Gu et al. 2017 292:12653).

Thus, there is compelling evidence that they truly report CBS site-specific effects.

Importantly, the FRET signal we measured correlates again with the observed mutant activities.

We agree with the Reviewer that CBS3 as the major CBS site is a confirmatory finding, and that effects of CBS1 and CBS4 mutants are original. At least the latter appears to have a structural function in allowing activation by ADP, most likely mediated by CBS3. Whether this structural function plays a regulatory role, e.g. depends on the adenylate bound to CBS4 and its potential exchange, cannot be answered by our data but remains an interesting hypothesis. We have revised the corresponding paragraph in our revised manuscript.

Line 194:

Our experiments clearly confirmed CBS3 as the main site of conformational response to saturating concentrations of both AMP and ADP (Fig. 3a)^{15, 51}.

8. Page 9 There is 2.0 mM MgCl₂ in the AMPfret purification buffers, was this removed before measuring the effect Mg²⁺ has on adenylate binding? This result indicates that if Mg²⁺ is added in excess in kinetic assays, then it wouldn't interfere with the other nucleotides (AMP or ADP) in the experiment. Thus, validating previous kinetic studies where Mg²⁺ is added in excess. Can the authors comment on this?

The Reviewer raises an important point that we now clarify in the revised manuscript. Namely, AMPK has been diluted from highly concentrated stock solution, so the effective Mg²⁺ concentration is below 20 microM and thus not relevant in the assays. In addition, we have added the data on ADP (Fig.4a), showing that there is no effect of Mg²⁺ as the Reviewer had already suspected. Thus, only ATP interaction with AMPK CBS sites is altered by complex formation with Mg²⁺, directly confirming earlier more indirect data cited by the Reviewer. We have added a comment in our revised manuscript:

Methods section (line 599):

Mg²⁺ effects on FRET were investigated with variable concentrations of Mg²⁺ (0 - 10 mM) and AMPfret stock solution diluted to yield final Mg²⁺ concentrations below 20 μM.

Results section (line 225):

The baseline FRET signal and the FRET increase induced by 20 μM AMP or 200 μM ADP were not sensitive to excess Mg²⁺ in our experiments, in agreement with structural studies performed with AMP in the presence or absence of Mg²⁺ ions⁵⁵.

Discussion section (line 380):

Our data compellingly demonstrate that not only for CBS binding^{24, 38}, but also for induced conformational changes the true competitor of AMP (and ADP) is non-complexed free ATP, which is at least 10-times less abundant in cells as compared to MgATP, while presence of Mg²⁺ does not alter AMP or ADP effects.

9. Page 10. Line 187: Our experiments clearly identify CBS3 as the main site of..... This is not a new finding, please change sentence accordingly and add appropriate references. Oakhill et al., 2010. PNAS. DOI:10.1073/pnas.1009705107; Chen et al., 2012. NSMB. DOI:10.1038/nsmb .The AMPfret construct provides an opportunity to test the relative contributions of AMP and ADP to the conformational change in the kinase. They could potentially answer the question where on the one hand Hardie's laboratory (ref 13) have argued AMP is the true activator whereas Oakhill and Carling have provided evidence for ADP being the major contributor.

As already stated in our response to the above Reviewer comment (7.), it was not our intention to claim the importance of CBS3 as a novel finding. Rather, our assay allows for the first time a direct confirmation of this feature, since it can be carried out without ATP (which in our hands always contains non-negligible amounts of both AMP and ADP as we determined by analytical methods). As to the dispute on the relative importance of AMP or ADP, our data support a contribution of both adenylates. Indeed, FRET occurs in a concentration range characteristic for AMP and ADP shifts at the onset of cellular energy stress. We have revised the discussion section in this respect:

Lines 420ff:

In vivo, ADP and AMP signals would occur together, with ATP consumption immediately increasing ADP, and - via the adenylate kinase reaction - also AMP^{9, 72}. Indeed, our data are consistent with both adenylates being important for AMPK activation in vivo, with activating conformational changes occurring at physiologically relevant concentrations, in particular very low micromolar AMP levels, and by using physiological adenylate mixtures.

10. Page 11. Is cellular resolution high enough to visualize specific localization of AMPK; for instance, AMPK accumulation on the lysosome as suggested in Zhang et al., 2017. Nature. 548 (112-116) DOI:10.1038/nature23275? If AMPfret can't localize to the nucleus, doesn't this invalidate AMPfret for localisation studies? i.e. AMPfret has altered protein-protein interactions blocking AMPK's normal activity in the nuclei, therefore other important protein-protein interaction might be blocked as well. Can the authors comment on this?

The Reviewer raises an interesting point. Indeed, we have looked for inhomogeneity within the cytosolic FRET signal but could not identify clear signals above noise level. This is probably not related to spatial resolution, but of the few AMPK complexes involved in these lysosomal signaling complexes. FRET sensors are likely not the method of choice for localization applications, and different, signal-amplifying detection techniques may be preferred. Otherwise, if AMPfret should be used to detect allosteric AMPK activation at specific subcellular sites, engineering of AMPfret variants with specific targeting signals would be likely required (e.g. Vishnu et al 2014 Mol Biol Cell 25:368). We suggest this approach also for improved detection of AMPfret inside the nucleus. We have reexamined nuclear AMPfret signals in a new series of experiments using HEK293 cells and could observe nuclear fluorescence clearly above background levels (albeit too low to reliably

determine FRET; see new Suppl. Fig. 7). Thus, nuclear import of AMPfret likely occurs but is ineffective, which could be remedied by adding nuclear localization signal (NLS) or deleting a nuclear export sequence (NES) identified by Kazgan et al. (2010 Mol Biol Cell 21:3433). At present, we can only speculate whether or not AMPfret preserves other interactions of native AMPK.

11. Page 12 Figure 5. In using the AMPfret construct in cells it would be important to know whether the construct itself disturbed the cell. What happens if a kinase dead version is used? It is not clear whether the AMPfret construct in cells works for drugs. Figure 5 the ACC loading blots have not come from the same gel as the pACC blots (band details do not match).

We observed that expression of the three AMPfret subunits (two tagged, one untagged) downregulates the levels of endogenous AMPK subunits. This ascertains that the large majority of AMPfret subunits recombines to yield fluorescent sensor, while only a small proportion recombines with endogenous AMPK to yield unproductive complexes (containing only one tagged fluorophore). Since endogenous AMPK does not disappear completely, and AMPfret has full AMPK activity/ligand affinities, we assume that AMPfret expression maintains AMPK signaling and does not generate a pronounced phenotype, consistent with our experimental observations. However, inactive AMPfret would downregulate AMPK activity with all potential cellular consequence, but we have not addressed this issue experimentally. Here, we did not test experimentally a range of pharmacological activators in cells, apart from our experiments with AICAR. For this type of screening applications, we suggest cell free assays that can be developed in a high throughput format. Controls for the ACC time course in Fig.5 were carried out with a gel run in parallel, which is now explained in the legend in our revised manuscript.

Line 1054:

AICAR-dependent activation of AMPK verified by parallel immunoblots for P-ACC and ACC (lower panel) and their quantification (upper panel).

12. Page 14 Line 320: AMPfret retains wild-type..... Authors should add here that in cells AMPfret doesn't translocate to the nucleus like wild-type.

As outlined in our response to the above Reviewer comment (10.), nuclear AMPfret import is not absent but just less effective (new Suppl. Fig. 7). We have added this information in our revised manuscript:

Line 345:

Nuclear import of AMPfret is reduced, but still occurring.

Overall this is a well written manuscript and the authors have provided strong support for their conclusions. However, the areas where their findings potentially contribute to a new understanding of AMPK are insufficiently developed.

- 1 They need to demonstrate the differential regulation by AMP and ADP at the CBS sites they have identified is not restricted to the mutant constructs they have used.
- 2 They need to answer the question of the relative contributions of AMP and ADP in modulating the conformation of AMPK
- 3 The general utility of the AMPfret construct is restricted to $\alpha 2\beta 2\gamma 1$ containing heterotrimers. This is a serious limitation for drug screening.

We agree with the Reviewer that our AMPfret sensor is a tool with great potential for many applications. In the present study, we provide comprehensive data-rich analysis of our sensor, providing unique new insight into the activation mechanisms of this AMPK complex. For other applications including screening procedures and cell-based assays, we provide compelling proof-of-concept.

We agree with the Reviewer that the data provided in our present study raises new and exciting questions that will be answered in future studies (probably then again raising entirely new questions). In the revised version of the manuscript, we have addressed the specific issues raised here by the Reviewer as follows:

1. These very same mutations at the CSB sites have been used in earlier publications to specifically delete the binding of allosteric activators to CBS3 (refs 15, 24), and yielded similar results as other sets of CBS mutations. These studies also showed that such mutations abolish AMPK activation by AMP without affecting AMPK catalytic activity in absence of AMP. To clarify, we have added in Methods:

Line 520ff:

Mutations at the $\alpha 2T172$ (T172D and T172A) and at three $\gamma 1$ CBS sites shown to abrogate adenylate binding^{17, 24} ...

2. A different presentation of the significance of CBS site mutants, including a comment on the importance of both AMP and ADP for AMPK activation:

Lines 420ff:

In vivo, ADP and AMP signals would occur together, with ATP consumption immediately increasing ADP, and - via the adenylate kinase reaction - also AMP^{9, 72}. Indeed, our data are consistent with both adenylates being important for AMPK activation in vivo, with activating conformational changes occurring at physiologically relevant concentrations, in particular very low micromolar AMP levels, and by using physiological adenylate mixtures.

3. We have the possibility, based on our ACEMBL cloning/recombination system, to generate other AMPK complexes for specific applications in future studies (e.g. including beta1 for drug screening or other isoform combinations for cellular applications that correspond to the major endogenously expressed isoforms; see lines 481ff).

Reviewer #4 (Remarks to the Author):

The authors have developed an interesting and original tool, AMPfret, for the direct study of AMPK conformational changes upon binding of nucleotides. They extensively characterize and validate AMPfret in vitro. Furthermore, they use AMPfret for a set of in vitro experiments that have an important impact on the understanding of the biochemical and physiological function of AMPK. An important advantage of genetically codified FRET sensors, as AMPfret, is the capacity of performing experiment in living cells. The set of AMPfret experiments in living cells here presented is limited in its scope.

We thank this Reviewer for this positive evaluation of our study.

Please find here below my specific comments:

1. Statistical analysis. The authors report in the Materials and Methods sections (lines 625 and 626) that 'where not stated otherwise, statistical significance was calculated by a two-tailed Student's t-test'. Requirements for the applicability of the t-test are that data are normally distributed and that are sampled from populations with equal variances. The authors do not report the results for tests of normality of distribution and homoscedasticity. Furthermore, in several instances multiple comparisons are performed (for example Fig.2, panel E; Fig.3, panel A; Fig.4, panels A and B; and Fig. 5, panel F). Multiple comparisons require an appropriate correction such as Bonferroni. If data are normally distributed, ANOVA should be first taken in consideration, followed by appropriate post hoc tests.

We agree with the Reviewer and have revised our manuscript accordingly. We have now tested all data for normality (Shapiro-Wilk) and equal variance. Data were further analyzed for significant differences by using one-way or two-way ANOVA, followed by post-hoc multiple pairwise comparisons by either Bonferroni or Student-Newman-Keuls methods, depending on the underlying data distribution as suggested by the software. This is detailed now in the Methods section (line 673) and in the relevant Figure Legends:

Data are presented as mean \pm standard error of the mean (SEM) and statistically analyzed by Sigma Plot 11.0 (Systat Software Inc., San José, CA, USA). Data were checked for normality (Shapiro-Wilk) and equal variance. If not stated otherwise, data were then analyzed for significance by one-way or two-way ANOVA, depending on the experimental design, followed by post-hoc multiple pairwise comparison by either Bonferroni or Student-Newman-Keuls methods. Differences were considered significant when $p \leq 0,02$. In the figures, means sharing the same letter do not differ significantly.

2. Lines 197 and 198. In order to assess proper agreement between the results obtained with AMPK activity assays of the CBS mutants and AMP-induced FRET changes, FRET experiments should be conducted in presence of all the concentrations of AMP and ADP used in the AMPK activity assays. Currently, FRET results, as reported in Fig.3 panel A, are only for 30microM AMP and 200microM ADP.

In this experiment, WT and CBS mutants can be readily compared at v_{max} with saturating concentrations of AMP (30 uM) and ADP (200 uM) that we use throughout our study. Correlation of FRET signals with AMPK activation is shown for WT in Fig.2A, and we do not repeat this type of assay here for CBS mutants. Note that exact correlation of

concentration dependencies will be limited due to ATP addition in kinase assays which introduce AMP and ADP contaminations as well as ATP competition at CBS sites.

3. Line 228. The authors write that they have used AMPfret 2.1 for the experiments reported in Fig.4, panel B. The figure legend (line 900) reports AMPfret 2.0. Also, Is AMPfret 1.0 been used for obtaining the results shown in Fig.4, panel A? This has not been specified either in the text or in the figure legend.

This has been corrected in the revised version of our manuscript.

4. Experiment in living cells. The authors report in the Materials and Methods section (lines 603 to 613) that an incubation chamber has been used for the confocal FRET experiments. Please specify the temperature at which the confocal experiments have been conducted as well the composition of the medium.

We have added the required information on culture conditions in the Methods section as follows:

Line 644:
37°C, 5% CO₂

Line 628ff:
3T3-L1 and HeLa cells were cultured in Dulbecco Modified Eagle Medium (DMEM, Institut de Biotechnologies Jacques Boy) containing 4,5 g/L glucose, 10% SVF (Pan-Biotech, lot P291905), 10 mM Hepes (PAA cell culture company), penicillin and streptomycin (Gibco) as well as non-essential amino acids (Sigma). HEK293t cells (ATCC) were cultured in DMEM (Gibco 41965039) containing 4,5 g/L glucose, 10% SVF, penicillin and streptomycin.

Line 640ff:
For AMPfret analysis, cells cultured in 8-well Labtek (Nunc) in complete medium were observed by confocal microscopy. Medium was then replaced by complete medium containing AICAR (1 mM final) or 2-DG (3.3g/L 2-DG and 1g/L glucose) without moving the 8-well plates.

Further acquisition parameters are required such as power of the excitation laser line, image pixel resolution, scanning frequency and temporal image resolution. The authors write that the signal of CFP and YFP fluorescences have been extracted and the FRET signal per single cells calculated. How?

We have added the required information on image acquisition and confocal data treatment in the Methods section as follows:

Line 645ff:
The instrument's autofocus system was used (LCS SP8) or z-stacks (4 tiles distributed through 10 µm) were acquired and combined (LCS SP2) to avoid focal drift during the experiments and to tolerate cell shape remodelling. For excitation, the 458 nm Argon laser (<20 % power) was used and pictures were taken at 512x512 or 1024x1024 pixel resolution with a 400 Hz scanning frequency and 4 lines averaging in the CFP (478±5 nm) and the YFP (530±5 nm) channels every 5 min for 45 min using HyD (LCS SP8) or PMT (LCS SP2) detectors. After averaging of z-stacks intensities and background subtraction (rolling ball radius: 200 pixels) using FIJI, both CFP and YFP fluorescence intensity values were

extracted from the recorded pictures using FIJI. FRET signal variations for single cells were calculated from averaged fluorescence intensities (YFP fluorescence value/CFP fluorescence value).

What is the FRET ratio plotted in Figure 5, panels C and E? Please, define it.

The FRET ratios plotted in Fig. 5 are means of FRET signals from individual cells after having used their values at $t=0$ for normalization to 1.

5. Please add dimension bars in Fig.5 panels A and D.

Dimension bars have been added.

6. Lines 259, 260 and Fig.5. There is minimal decrease of the fluorescence intensity of the donor upon AICAR exposure (Fig.5, panel B). This is concerning. Upon FRET, increase of the sensitised emission of the acceptor and decrease emission of the donor are expected. Can the authors provide the average values of the fluorescence intensities in the donor channel (478 \pm 10nm) and acceptor channel (530 \pm 10nm) before and upon exposure to AICAR for the experiments plotted in panel C and E?

We have repeated wavelength scans in a new series of experiments with HEK293 cells (new Fig. 5b). It seems that fluorescence emission in the range of 470-510 nm is disturbed in the cellular background, not yielding ideal spectra as compared to purified AMPfret protein complex. However, we do see a decrease in average fluorescence in the CFP channel (468-488 nm) as compared to an increase in the YFP channel 520-540).

7. Are the n values in the legend of Fig.5, panel C and E referring to the number of single analysed cells? If so, how many independent experiments have been performed for the data shown in panel C and E?

We agree that the information was incomplete and now specify in our revised manuscript as follows:

Fig 5c (line 1044):

Data are mean \pm SEM (n=64, n=27 and n=39 individual cells monitored through at least two independent experiments for 2-DG, AICAR and Ctl conditions, respectively).

Fig5f (line 1053):

Data are mean \pm SEM (n=9 cells of two independent experiments).

Fig5i (line 1061):

Data are mean \pm SEM (n=51 cells of three independent experiments).

8. The behaviour of AMPfret 2.1 in living cells has been assessed by exposure to the AMPK activator AICAR. This is interesting but it does not allow comparison to the in vitro behaviour since 991 was used instead. An experiment in living cells with exposure to 991 should be conducted. Furthermore, the capacity of AMPfret 2.0 of reporting changes of AMPK activity should also be assessed by inhibition of mitochondria respiration.

We have now carried out additional experiments exposing HEK293 cells to either AICAR or 2-deoxyglucose (2-DG). 2-DG induces energy stress by decreasing ATP/ADP and ATP/AMP ratios (new Fig.5d). Consistent with what would be expected, AICAR increases AMPfret signals faster than 2-DG, since its cellular products directly act on AMPK (new Fig.5c). However, with 2-DG much higher AMPfret signals can be reached, since here not only does AMP increase (like ZMP with AICAR), but at the same time ATP decreases drastically. These data indicate that AMPfret can properly report cellular energy states. We present the new data in a novel paragraph in the revised Methods section (lines 632ff) and Results section (lines 285ff). Moreover, we added a comment in the Discussion in our revised manuscript (line 456ff). These changes have been already detailed above in our responses to Reviewer comments (comment 4 of reviewer 1). Concerning the comparison of AMPK activators *in vitro* and in cells, we argue that the effect of AMP in the cell-free system can be compared to AICAR in the cellular systems, since the finally active compounds of AICAR are the AMP-analogue ZMP as well as AMP itself.

We thank all Reviewers for their insightful comments which we have addressed in full, and which helped us improve our manuscript. We are hopeful that the revised version of our manuscript will be accepted for publication in *Nature Communications*.

Reviewers' Comments:

Reviewer #1:

Remarks to the Author:

The authors have answered most of the points I raised on the first version in a satisfactory manner. I have two remaining points:

MAJOR POINT:

1. Partly in response to my original point 4, the authors have added new data obtained using 2-deoxyglucose. While this is indeed worthwhile, it is a little disappointing that they only show data obtained for AMP/ATP and ADP/ATP ratios at a single time point (new Fig. 5d), rather than attempting to see whether changes in these ratios correlated with changes in the FRET ratio (Fig. 5c).

MINOR POINT:

2. Line 96: the authors have indeed studied the behaviour of their FRET sensor in "living cells", but most researchers would not consider cultured cells grown in 2-D to be "in vivo". I suggest that the latter two words are simply deleted.

Reviewer #2:

Remarks to the Author:

The revised manuscript from Pelosse et al is much improved and all of my questions and concerns have been addressed to my satisfaction. In particular, it was useful to see AMPfret experiments performed with a kinase inhibitor as well as rough quantitation of changes in nucleotide levels in experiments with HEK393 cells.

In addition to new experimental data (addressing multiple reviewer comments), I believe that text and methods have been clarified at multiple places.

This work remains an interesting study for all those in the AMPK field.

Reviewer #3:

Remarks to the Author:

The authors have adequately responded to all Referee 3's comments. However I remain concerned about the general utility of the AMPfret system beyond the highly AMP-responsive $\alpha 2\beta 2\gamma 1$ AMPK construct to the other AMPK isoforms.

Reviewer #4:

Remarks to the Author:

The authors have revised the manuscript for statistical analysis as required.

I have appreciated the effort of the authors in addressing further the characterization of the AMPfret sensor in intact cells by adding experiments with 2-DG.

Line 290-291: the authors report as data not shown that AMPfret reaches saturation for time >50 min. These data should be shown in figure 5, panel C.

Line 475-476. The authors report as data not shown that basal activation of AMPK is increased after transient transfection. The basal level of AMPK activation after transfection should be reported as supplementary material for all the cell lines studied.

REVIEWERS' COMMENTS:

Reviewer #1 (Remarks to the Author):

The authors have answered most of the points I raised on the first version in a satisfactory manner. I have two remaining points:

We thank for the positive re-evaluation of our revised manuscript.

MAJOR POINT:

1. Partly in response to my original point 4, the authors have added new data obtained using 2-deoxyglucose. While this is indeed worthwhile, it is a little disappointing that they only show data obtained for AMP/ATP and ADP/ATP ratios at a single time point (new Fig. 5d), rather than attempting to see whether changes in these ratios correlated with changes in the FRET ratio (Fig. 5c).

We agree with the reviewer that a parallel time course of FRET and adenylate ratios would be worthwhile. However, while confocal FRET analysis is done with few cells in 8-well LabTec plates, adenylate extraction and HPLC analysis (in particular for quantitative AMP detection) requires large volumes of cells pooled from several cell culture dishes. These different conditions make it difficult to realize exact temporal correlation of FRET and adenylate levels. Such cumbersome adenylate detection is in fact a key argument for using an endogenous sensor such as AMPfret. Thus, we have done only end point determination of adenylates, when kinetics approach steady state.

MINOR POINT:

2. Line 96: the authors have indeed studied the behaviour of their FRET sensor in "living cells", but most researchers would not consider cultured cells grown in 2-D to be "in vivo". I suggest that the latter two words are simply deleted.

We agree and have deleted these words on p.4.

Reviewer #2 (Remarks to the Author):

The revised manuscript from Pelosse et al is much improved and all of my questions and concerns have been addressed to my satisfaction. In particular, it was useful to see AMPfret experiments performed with a kinase inhibitor as well as rough quantitation of changes in nucleotide levels in experiments with HEK393 cells.

In addition to new experimental data (addressing multiple reviewer comments), I believe that text and methods have been clarified at multiple places.

This work remains an interesting study for all those in the AMPK field.

We thank for the positive re-evaluation of our revised manuscript, and are happy that we could satisfy all concerns of the reviewer.

Reviewer #3 (Remarks to the Author):

The authors have adequately responded to all Referee 3's comments. However I remain concerned about the general utility of the AMPfret system beyond the highly AMP-responsive $\alpha 2\beta 2\gamma 1$ AMPK construct to the other AMPK isoforms.

We thank for the positive re-evaluation of our revised manuscript, and are happy that we could satisfy all concerns of the reviewer.

Reviewer #4 (Remarks to the Author):

The authors have revised the manuscript for statistical analysis as required. I have appreciated the effort of the authors in addressing further the characterization of the AMPfret sensor in intact cells by adding experiments with 2-DG.

We thank for the positive re-evaluation of our revised manuscript.

Line 290-291: the authors report as data not shown that AMPfret reaches saturation for time >50 min. These data should be shown in figure 5, panel C.

Since we do not have data points for control and AICAR beyond 45 min, we prefer to maintain the figure as it is. Instead, we have reformulated the text, just stating: "At the end of the 45 min observation period, the FRET signal was already about twice as high as with AICAR, without yet reaching saturation." (p.39)

Line 475-476. The authors report as data not shown that basal activation of AMPK is increased after transient transfection. The basal level of AMPK activation after transfection should be reported as supplementary material for all the cell lines studied.

We have deleted the corresponding sentence referring to data not shown. At present, we do not have data for all used cell lines with sufficient statistical power to be included in the ms. However, preliminary results with HeLa cells clearly suggest an effect of the chemical transfection procedure on energy state and AMPK activation, which we will examine in a follow-up study.